# RESOURCE-EFFICIENT MODEL-FREE REINFORCEMENT LEARNING FOR BOARD GAMES

## ABSTRACT

Board games have long served as complex decision-making benchmarks in artificial intelligence. In this field, search-based reinforcement learning methods such as AlphaZero have achieved remarkable success. However, their significant computational demands have been pointed out as barriers to their reproducibility. In this study, we propose a model-free reinforcement learning algorithm designed for board games to achieve more efficient learning. To validate the efficiency of the proposed method, we conducted comprehensive experiments on five board games: Animal Shogi, Gardner Chess, Go, Hex, and Othello. The results demonstrate that the proposed method achieves more efficient learning than existing methods across these environments. In addition, our extensive ablation study shows the importance of core techniques used in the proposed method. We believe that our efficient algorithm shows the potential of model-free reinforcement learning in domains traditionally dominated by search-based methods.

## 1 INTRODUCTION

Board games are highly complex decision-making tasks as even human experts may spend minutes to hours deliberating on a single action. Due to this complexity, they have served as a canonical benchmark for artificial intelligence for several decades (Samuel, 1959; Tesauro et al., 1995; Campbell et al., 2002; Silver et al., 2016).

In this field, search-based methods such as AlphaZero (Silver et al., 2018) have achieved a milestone by learning to play Chess, Shogi, and Go by a single reinforcement learning (RL) algorithm. Due to its generality and independence from game-specific knowledge, it has also been applied to various fields beyond board games (Schrittwieser et al., 2020; Hubert et al., 2021; Fawzi et al., 2022; Mankowitz et al., 2023).

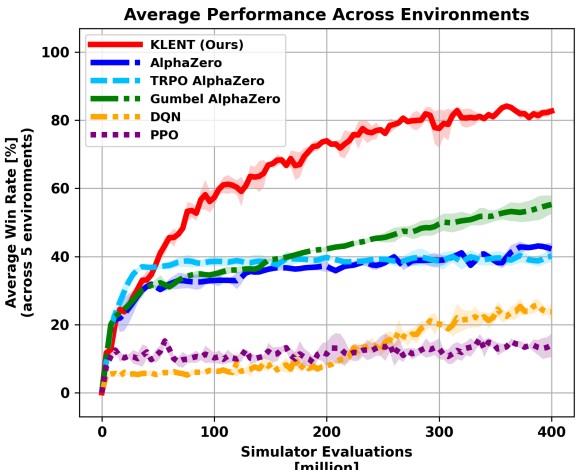

Figure 1: Average performance across the five board games. The proposed method (KLENT) achieves efficient learning compared to existing approaches.

While the success of search-based methods in the board game domain is remarkable, their substantial computational demands are often cited as a barrier to their reproducibility (Zhao et al., 2022). Indeed, it has been reported that the training process of AlphaZero requires more than 10 GPU-years (Silver et al., 2018; Tian et al., 2019). To address this issue, recent studies have proposed modified search-based algorithms that reduce the depth and rollout count of the search tree (Hessel et al., 2021; Danihelka et al., 2022), based on the observation that most of the computational cost is incurred during sample collection with Monte-Carlo tree search (MCTS). Although these methods lessen the reliance on look-ahead search during training, it remains an open question whether that reliance can be entirely eliminated.

In other fields of RL such as robotics, model-free methods tend to be preferred due to their implementation simplicity and computational efficiency (Kroemer et al., 2021; Tang et al., 2025). Nonetheless, the effectiveness of model-free RL in the board game domain has not been thoroughly investigated.

These situations raise the following question: *Can we design a model-free RL algorithm for board games that can learn a competitive policy with fewer training resources?* In this paper, we propose a model-free reinforcement learning algorithm that completely eliminates look-ahead search during training. The proposed method is based on a policy optimization approach which we refer to as Kullback-Leibler and Entropy Regularized Policy Optimization (KLENT). To validate the efficiency of the proposed method, we have conducted comprehensive experiments on five board games, namely Animal Shogi, Gardner Chess, Go, Hex, and Othello. As shown in Figure 1, our method achieves efficient learning compared to existing methods.

The proposed method KLENT incorporates three techniques from the field of RL: Kullback-Leibler (KL) regularization for gradual policy updates, entropy regularization for encouraging exploration, and $\lambda$-returns for efficient value function learning. Through an extensive ablation study, we empirically demonstrate that the combination of these three techniques is essential for efficient learning.

> The main contributions of this work are summarized as follows:
>
> 1. We propose a model-free reinforcement learning algorithm designed for board games, which we refer to as KLENT.
> 2. Through comprehensive experiments, we demonstrate that KLENT achieves more efficient learning than existing methods.
> 3. We conduct an extensive ablation study, which validates that three components of KLENT, namely KL regularization, entropy regularization, and $\lambda$-returns, are essential for its efficiency.

We have demonstrated that properly revisiting existing techniques can achieve efficient learning in the board game domain, and also shown that the proper combination is important to achieve high performance. We consider the novelty of this study lies in our empirical result, as search-based methods such as AlphaZero have been believed to be state-of-the-art in this domain.

In this study, we focus on "board games", but as we define in Section 2.2, we only assumed that the environment is an MDP with a finite action space. This class of problems covers several real-world applications such as discrete optimization, algorithmic discovery, and mathematical proving (Fawzi et al., 2022; Mankowitz et al., 2023; Hubert et al., 2025) and search-based methods such as AlphaZero are widely used in this domain. Our efficient algorithm may also achieve efficient learning in these practical domains, accelerating research on such real-world problems.

## 2 PROBLEM SETTING

### 2.1 REINFORCEMENT LEARNING

In this study, we formulate board games as reinforcement learning problems. Reinforcement learning (RL) (Sutton et al., 1998) is a framework in which an agent learns a policy $\pi$ through interactions with an environment to maximize an expected return. This framework can be formalized as a Markov Decision Process (MDP) (Bellman, 1957), consisting of a state space $\mathcal{S}$, an action space $\mathcal{A}$, a transition probability function $P(s'|s,a)$, a reward function $r(s,a)$, and a discount factor $\gamma \in [0,1]$. At each time step $t$, the agent selects an action $A_t \in \mathcal{A}$ based on its policy $\pi(A_t|S_t)$ and the current state $S_t \in \mathcal{S}$. In response, the environment transitions to the next state $S_{t+1} \in \mathcal{S}$ according to the transition probability $P(S_{t+1}|S_t, A_t)$ and provides a reward $R_t = r(S_t, A_t)$. The objective of the agent is to maximize the expected return $\mathbb{E}_{(S_t, A_t, R_t) \sim (P,\pi)} \left[ \sum_{t=0}^{T} \gamma^t R_t \right]$. Here, $T$ represents the terminal timestep of an episode. The state-value function $V^\pi(s) = \mathbb{E}_{(P,\pi)}[\sum_{t=0}^{T} \gamma^t R_t | S_0 = s]$ and the action-value function $Q^\pi(s,a) = \mathbb{E}_{(P,\pi)}[\sum_{t=0}^{T} \gamma^t R_t | S_0 = s, A_0 = a]$ can be used to evaluate and improve the policy $\pi$.

## 2.2 BOARD GAMES

Board games have long been used as important benchmarks for artificial intelligence (Samuel, 1959; Tesauro et al., 1995; Campbell et al., 2002; Silver et al., 2016; Yannakakis & Togelius, 2018). In this study, the term "board games" is used specifically to indicate perfect-information games with a finite action space, such as the game of Go. This kind of games can be formulated as an MDP, with the reward assigning $R_T = +1$ for a win, $R_T = -1$ for a loss, and $R_T = 0$ for a draw, while rewards are zero at all other timesteps $t \in \{0, 1, \ldots, T-1\}$. By setting the discount factor $\gamma$ to 1, we ensure that the final outcome of the game is directly reflected in the expected return.

## 3 RELATED WORK

### 3.1 MODEL-FREE APPROACHES

Classical RL algorithms include TD-learning (Sutton, 1988b) and Q-learning (Watkins & Dayan, 1992). DQN (Mnih et al., 2015) combined deep learning and Q-learning and succeeded in Atari domain. Several RL algorithms have been proposed based on the paradigm of regularized policy optimization, which can generally be formulated as follows:

$$\underset{\pi'}{\text{maximize}} \; \mathbb{E}_{A \sim \pi'(\cdot|s)}[Q^\pi(s, A)] - \mathcal{R}(\pi'). \tag{1}$$

Here, $\pi'$ is the optimized policy, $\pi$ is the prior policy, and $\mathcal{R}(\pi')$ is the regularization term. For example, if we define the regularization term as $\mathcal{R}(\pi') = -\alpha H(\pi')$, where $H(\pi')$ is the entropy of the optimized policy $\pi'$, the optimal solution corresponds to a softmax policy $\pi(a|s) \propto \exp(Q^\pi(s, a)/\alpha)$. This policy has long been adopted in prior studies, including classical approaches such as REINFORCE (Williams, 1992) and SARSA (Rummery & Niranjan, 1994; Van Seijen et al., 2009). In addition, Soft Q-Learning (Haarnoja et al., 2017) and SAC (Haarnoja et al., 2018) are methods that treat the entropy term as additional rewards.

Alternatively, if we define the regularization term as the difference between prior policy $\pi$ and optimized policy $\pi'$, we can make the policy updates gradual. For example, TRPO (Schulman et al., 2015) and one variant of PPO (Schulman et al., 2017) use the forward KL divergence $\mathcal{R}(\pi') = \beta D_{\text{KL}}(\pi \| \pi')$ and MPO (Abdolmaleki et al., 2018) use the reverse KL divergence $\mathcal{R}(\pi') = \beta D_{\text{KL}}(\pi' \| \pi)$. Entropy regularization can also be combined to enhance exploration for these methods. Interestingly, Grill et al. (2020) have pointed out that AlphaZero (Silver et al., 2018) is also approximately solving a policy optimization problem with KL regularization.

For the methods that leverage both reverse KL regularization and entropy regularization, Vieillard et al. (2020), Sokota et al. (2022), and Zhan et al. (2023) provided the theoretical properties of this combination. In this study, we aim to empirically show that this combination is effective for achieving efficient learning in the board game domain.

Some of the model-free approaches are tested in the board game domain. For example, Schraudolph et al. (1993), Thrun (1994), and Tesauro et al. (1995) have tried TD-learning in Go, Chess, and Backgammon, respectively, and Van Der Ree & Wiering (2013) have tested TD-learning, Q-learning, and SARSA in Othello. However, search-based approaches are considered to be stronger in this domain nowadays, which we explain in the next section.

### 3.2 SEARCH-BASED APPROACHES

Search-based approaches have demonstrated strong performance in board games. Classical approaches include TD-leaf($\lambda$) (Baxter et al., 1998), which combines TD-learning with look-ahead search. One of the most well-known algorithms is AlphaGo (Silver et al., 2016). It combined supervised pre-training with human expert game records and fine-tuning by RL with MCTS, defeating a human world champion in the game of Go. AlphaGo Zero (Silver et al., 2017) eliminated the need for supervised pre-training, and AlphaZero (Silver et al., 2018) extended it to general perfect-information finite-action games. Its generality has enabled applications in other fields, including mathematical and algorithmic discovery (Fawzi et al., 2022; Mankowitz et al., 2023). Subsequent studies of AlphaZero (Schrittwieser et al., 2020; Hubert et al., 2021; Ozair et al., 2021; Schrittwieser et al., 2021) have extended its applicability to a wider range of RL settings, such as continuous action spaces and partial observations.

While these search-based approaches are powerful, their significant computational demand has been pointed out as limitations (Zhao et al., 2022). To address this issue, recent studies have proposed more lightweight and efficient search-based algorithms. For example, Hessel et al. (2021) proposed a method that reduces the depth of the search tree and performs a one-step search instead of a deep MCTS. Furthermore, Danihelka et al. (2022) proposed Gumbel AlphaZero, which reduces the rollout count of tree search, achieving efficient learning in the board game domain. Our work shares the goal of achieving efficient learning with these prior studies, but takes a more drastic approach. While previous methods reduce the amount of look-ahead search during training, we aim to completely eliminate it. In this sense, our method can be regarded as the zero-search limit of this line of research.

### 3.3 Game-Specialized Approaches

Another line of research aims to enhance game-playing agents by incorporating game-specific knowledge. This approach has been adopted in both perfect-information games (Romstad et al., 2016; Delorme, 2017; Wu et al., 2020) and imperfect-information games (Moravčík et al., 2017; Li et al., 2020; Perolat et al., 2022; Bakhtin et al., 2023), leading to strong performance. However, our aim is to design a game-agnostic pure reinforcement learning method for board games, which distinguishes our work from these prior studies.

## 4 KLENT: KL AND ENTROPY REGULARIZED POLICY OPTIMIZATION

In this study, we propose Kullback-Leibler and Entropy Regularized Policy Optimization (KLENT). KLENT is an on-policy model-free RL algorithm, which is designed to achieve efficient learning in the board game domain. In Section 4.1, we describe our policy update rule, detailing our policy optimization problem and the solution to it. In Section 4.2, we explain value function learning methodology, utilizing $\lambda$-returns to stabilize the learning process. Section 4.3 presents the overall algorithm of KLENT with a pseudo code.

### 4.1 Policy Update Rule

To design the policy update rule, we employ the paradigm of regularized policy optimization. To avoid abrupt policy changes and achieve gradual policy updates, we utilize reverse KL regularization. In addition, to maintain sufficient exploration to ensure sample diversity and prepare for unknown opponents, we also incorporate entropy regularization. Using these two regularizers, we consider the following regularized policy optimization problem.

$$\underset{\pi'}{\text{maximize}} \; \mathbb{E}_{A \sim \pi'(\cdot|s)}[Q^\pi(s, A)] - \beta D_{\text{KL}}(\pi'(\cdot|s) \| \pi(\cdot|s)) + \alpha H(\pi'(\cdot|s)). \quad (2)$$

Here, $D_{\text{KL}}(\pi'\|\pi)$ is the reverse KL divergence between the new policy $\pi'$ and the current policy $\pi$, and $H(\pi')$ is the entropy of $\pi'$. The coefficients $\alpha$ and $\beta$ are the non-negative scalar hyperparameters which control the strength of the regularization terms. Leveraging the fact that the action space $\mathcal{A}$ of board games is finite, the optimal solution $\pi'$ can be analytically derived in the following closed-form expression:

$$\pi'(a|s) = \frac{1}{Z(s)} \exp\left(\frac{Q^\pi(s, a) + \beta \log \pi(a|s)}{\alpha + \beta}\right), \quad (3)$$

where $Z(s) = \sum_{a \in \mathcal{A}} \exp\left(\frac{Q^\pi(s,a) + \beta \log \pi(a|s)}{\alpha + \beta}\right)$ is a normalization term to ensure that $\pi'(\cdot|s)$ is a probability distribution. Appendix A provides the detailed derivation of this optimal solution. In KLENT, this analytically obtained policy $\pi'$ is used for action selection during the training.

We model the policy as $\pi_\theta(a|s)$ with a neural network. When updating the parameter $\theta$, the analytically obtained optimal policy $\pi'(\cdot|s)$ is used as the learning target, and fitting of $\theta$ is conducted to minimize the cross-entropy $-\sum_{a \in \mathcal{A}} \pi'(a|s) \log \pi_\theta(a|s)$.

**Algorithmic differences from prior studies** The idea of KL and entropy regularization is shared with Sokota et al. (2022) and Perolat et al. (2021). The main difference of KLENT from them lies in the use of Equation 3. KLENT uses the analytically obtained solution for the learning target,

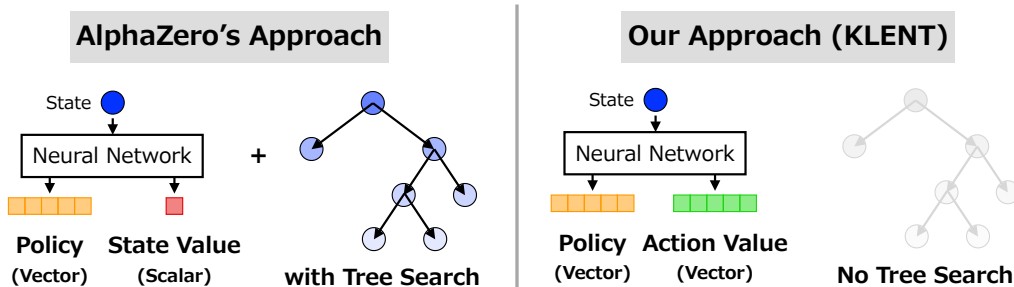

Figure 2: Conceptual comparison between AlphaZero and the proposed method KLENT. Search-based methods such as AlphaZero model the policy and the value-function $V(s)$, and use tree search to estimate the Q-function $Q(s, a)$. KLENT, by contrast, directly models both the policy and the Q-function using neural networks, eliminating the need for search.

while prior studies used PPO-style policy update and replicator dynamics, respectively. The idea of entropy regularization is also shared with SAC (Haarnoja et al., 2018). The main differences between SAC and KLENT lie in action spaces, value learning, and on-policy-ness. SAC focuses on continuous action spaces, soft Q-function, and off-policy settings. In contrast, KLENT focuses on finite action spaces, ordinary Q-function, and on-policy settings. The main difference between Grill et al. (2020) and KLENT is that while the prior study uses MCTS for action-value estimation and uses forward KL regularization, KLENT eliminates MCTS and uses reverse KL regularization.

## 4.2 LEARNING ACTION-VALUE FUNCTION

To compute the optimized policy $\pi'(a|s)$ in Equation 3, the probability given by the prior policy $\pi(a|s)$ and the estimate of the action value $Q^\pi(s, a)$ are required for all actions $a \in \mathcal{A}$. In search-based methods such as AlphaZero, the state-value function $V^\pi(s)$ is modeled, and the action value is estimated from backup values obtained through tree search. By contrast, because our goal is to develop a model-free approach without look-ahead search, such backup values are unavailable. To bridge this gap, we directly model the action-value function as $Q_\theta(s, a)$, instead of the state-value function. This conceptual difference is illustrated in Figure 2.

This choice of directly modeling the action-value function can make value-function learning more challenging, because learning the action-value function is often harder than learning the state-value function because of the following reasons. First, while the state-value function is a mapping $V^\pi : \mathcal{S} \to \mathbb{R}$, the action-value function $Q^\pi : \mathcal{S} \times \mathcal{A} \to \mathbb{R}$ must capture action-conditional variation at every state. As $|\mathcal{A}|$ grows, the resulting function class expands substantially, making the action-value function harder to learn.

In addition, learning the action-value function often requires reasoning over more complex spatial features. For example, Figure 3 illustrates a position in which Black places a stone at the location marked with a green triangle, capturing 16 white stones indicated by red crosses. In such positions, recognizing that this move has high action value depends on interpreting intricate spatial configuration of surrounding stones, whereas the state value is often estimable from simpler features such as stone counts. Within these difficulties, learning the action-value function require additional care.

To achieve stable and efficient action-value learning, we consider the use of $\lambda$-returns (Sutton, 1988a) is an effective approach. For $\lambda \in [0, 1]$, $n$-step return $G_t^{(n)}$ and $\lambda$-return $G_t^\lambda$ are defined as follows:

$$G_t^{(n)} = \sum_{k=0}^{n-1} \gamma^k R_{t+k} + \gamma^n \hat{v}_{t+n}, \qquad G_t^\lambda = (1 - \lambda) \sum_{n=1}^{T-t-1} \lambda^{n-1} G_t^{(n)} + \lambda^{T-t-1} G_t^{(T-t)}. \quad (4)$$

Here, $\hat{v}_{t+n}$ denotes a bootstrap estimate of the state value $V^\pi(S_{t+n})$. In general, the benefit of $\lambda$-returns can be explained in terms of the bias-variance trade-off. The bias-variance decomposition is given by the following formula:

$$\underbrace{\mathbb{E}\big[(\hat{X} - x)^2\big]}_{\text{Mean Squared Error}} = \underbrace{\big(\mathbb{E}[\hat{X}] - x\big)^2}_{\text{Squared Bias}} + \underbrace{\mathrm{Var}(\hat{X})}_{\text{Variance}}, \quad (5)$$

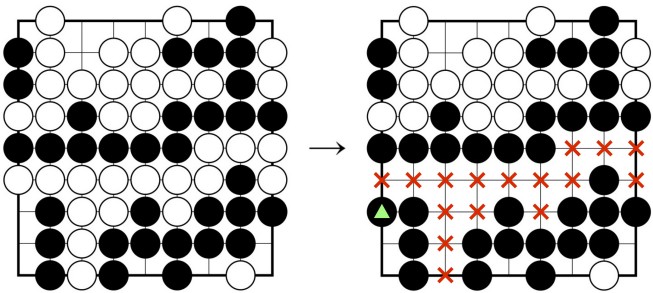 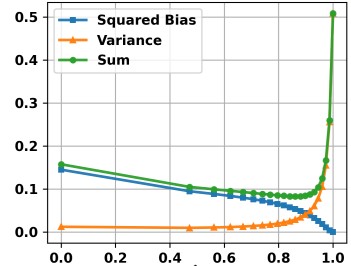

Figure 3: Illustration of difficulty in learning action-value in 9x9 Go. Learning the action-value function is generally more difficult than learning the state-value function, as it often requires handling more complex spatial features.

Figure 4: Bias-Variance tradeoff in 9x9 Go. Intermediate $\lambda$ minimizes the sum of squared bias and variance.

where the estimand is the true action value, $x = Q^\pi(S_t, A_t)$, and the estimator is the $\lambda$-return, $\hat{X} = G_t^\lambda$ in our case. We empirically validated this bias-variance trade-off through a preliminary experiment. We measured the squared bias, the variance, and their sum of the $\lambda$-return using a pretrained model provided by Koyamada et al. (2023) in 9x9 Go environment. The results in Figure 4 suggest that, in board games as well, increasing $\lambda$ reduces bias while increasing variance, with an intermediate value minimizing their sum.

Motivated by these observations, we employ the $\lambda$-return $G_t^\lambda$ as the learning target for the action-value $Q_\theta(S_t, A_t)$. We obtain the state-value estimate $\hat{v}_{t+n}$ by explicitly computing the expectation as follows.

$$\hat{v}_{t+n} = \mathbb{E}_{A \sim \pi'(\cdot|S_{t+n})}[Q_\theta(S_{t+n}, A)] = \sum_{a \in \mathcal{A}} \pi'(a|S_{t+n})Q_\theta(S_{t+n}, a). \tag{6}$$

We also empirically demonstrate through ablation experiments in which $\lambda$ is set to 0 or 1, corresponding to the 1-step return and the Monte Carlo return respectively, that an intermediate value of $\lambda$ contributes to learning efficiency. KLENT utilizes $\lambda$-returns, which are designed for on-policy settings. For extending KLENT to off-policy settings, it can be realized by replacing $\lambda$-returns off-policy counterparts, such as Retrace($\lambda$) (Munos et al., 2016).

### 4.3 OVERALL ALGORITHM

The overall procedure of the proposed algorithm KLENT is illustrated in Algorithm 1. Starting from randomly initialized networks, the proposed algorithm updates the policy $\pi_\theta$ and the action-value function $Q_\theta$ alternating a self-play phase for data collection and a fitting phase for network updates.

In the self-play phase, the goal is to populate the on-policy sample buffer $\mathcal{D}$. During the episode, the actions are sampled from the policy $\pi'$ using the current networks $\pi_\theta$ and $Q_\theta$. After the episode terminates, the $\lambda$-return $G_t^\lambda$ is computed for all timesteps $t$ by Equation 4. Samples are collected by repeatedly running episodes until the number of samples in the buffer reaches a predefined capacity.

---

**Algorithm 1** KLENT Algorithm

1: Initialize the policy network $\pi_\theta(a|s)$.
2: Initialize the action-value network $Q_\theta(s, a)$.
3: **repeat**
4:     $\mathcal{D} \leftarrow \{\}$                ▷ on-policy sample buffer
5:     **repeat**
6:         Initialize the state $S_0$.
7:         **for** $t = 0, \ldots, T$ **do**
8:             $\pi'(a|S_t) \propto \exp\left(\frac{Q_\theta(S_t, a) + \beta \log \pi_\theta(a|S_t)}{\alpha + \beta}\right)$
9:             $\hat{v}_t \leftarrow \mathbb{E}_{A \sim \pi'(\cdot|S_t)}\left[Q_\theta(S_t, A)\right]$
10:            Sample $A_t \sim \pi'(\cdot|S_t)$.
11:            Execute $A_t$ and observe $(S_{t+1}, R_t)$.
12:         **end for**
13:         Compute $\lambda$-returns $\{G_t^\lambda\}_{t=0}^T$ using equation 4.
14:         $\mathcal{D} \leftarrow \mathcal{D} \cup \left\{(S_t, A_t, (\pi'(a|S_t))_{a \in \mathcal{A}}, G_t^\lambda)\right\}_{t=0}^T$
15:     **until** $\mathcal{D}$ reaches a predefined capacity.
16:     Update $\theta$ by minimizing $L(\theta)$ in equation 7.
17: **until** convergence.

---

In the fitting phase, the data accumulated in the buffer $\mathcal{D}$ is used to update the network parameter $\theta$. The loss function $L(\theta)$ is defined as follows:

$$L(\theta) = \mathbb{E}_{\mathcal{D}}\left[ -\sum_{a \in \mathcal{A}} \pi'(a|S) \log \pi_\theta(a|S) + (Q_\theta(S, A) - G^\lambda)^2 \right]. \tag{7}$$

Here, $\mathbb{E}_{\mathcal{D}}[\cdot]$ indicates that $(S, A, (\pi'(a|S))_{a \in \mathcal{A}}, G^\lambda)$ are sampled from the buffer $\mathcal{D}$. This loss function is designed to simultaneously optimize the policy and action-value networks, with the analytically obtained policy $\pi'(\cdot|S)$ and $\lambda$-return $G^\lambda$ serving as targets for learning. By iterating these self-play and fitting phases, the policy $\pi_\theta$ and the action-value function $Q_\theta$ are progressively refined and eventually become strong.

## 5 EXPERIMENTS

In this section, we present our experimental results on board games. Specifically, Section 5.1 provides the results of performance comparison on five board games, demonstrating the efficiency of the proposed method KLENT compared to existing methods. Subsequently, we present the results of our ablation study in Section 5.2, demonstrating the importance of the key techniques in KLENT, namely KL regularization, entropy regularization, and $\lambda$-returns. Lastly, we provide the experimental results on large-scale 19x19 Go in Section 5.3. Our empirical analysis on the convergence limit of KLENT is also provided in Appendix L.

### 5.1 PERFORMANCE COMPARISON

We conducted experiments to compare the performance and the learning efficiency of KLENT and existing approaches. We employed five medium-scale board games, namely, Animal Shogi, Gardner Chess, 9x9 Go, Hex, and Othello as benchmark environments. The observation shape and action space size for each game are summarized in Table 1.

To assess the playing strength of each agent, we used pretrained checkpoints from the Pgx library (Koyamada et al., 2023) for anchored opponents and measured the win rates against them. To evaluate the learning efficiency, we employed the number of simulator evaluations on the horizontal axis, which corresponds to the number of neural network queries during self-play. This metric serves as an indicator of the computational demand of training processes and has also been adopted in the literature, particularly when training efficiency is of the primary interest (Wu et al., 2020). For search-based methods, the number of data collected is the number of simulator evaluations divided by the number of MCTS simulations. For model-free methods, the number of data collected is equal to the number of simulator evaluations.

As baselines for performance comparison, we used AlphaZero (Silver et al., 2018), TRPO AlphaZero (Grill et al., 2020), and Gumbel AlphaZero (Danihelka et al., 2022) as search-based approaches, and DQN (Mnih et al., 2015) and PPO (Schulman et al., 2017) as model-free approaches. The network architecture was unified across all experiments. Specifically, a ResNet (He et al., 2016) with 6 residual blocks was used for feature extraction. Depending on the method, additional heads such as a policy head, an action-value head, and a state-value head were added. These heads were designed as multilayer perceptrons. Further details of the experimental setup and implementation are provided in Appendix B and Appendix C, respectively.

Table 1: Five board game environments used for the experiments.

| Game Name | Animal Shogi | Gardner Chess | 9x9 Go | Hex | Othello |
|---|---|---|---|---|---|
| **Initial State** |  |  |  |  |  |
| **Observation Shape** | (4, 3, 194) | (5, 5, 115) | (9, 9, 17) | (11, 11, 4) | (8, 8, 2) |
| **Action Space Size** | 132 | 1225 | 82 | 122 | 65 |

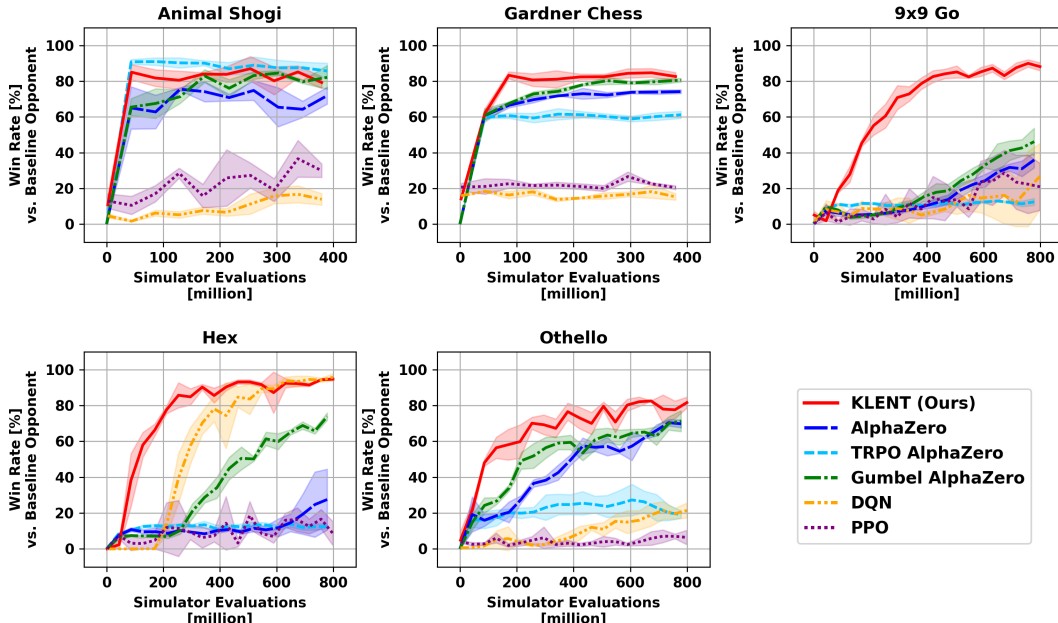

Figure 5: Performance comparison between the proposed method KLENT and existing methods. KLENT achieves competitive or higher efficiency compared to existing methods.

In the evaluation, we have used a reactive policy in our main experiments and provided results with test-time search in Appendix H. We have compared performances by fixing the test-time simulation count for each method. As we use the same ResNet architecture for each method, this is also equivalent to fixing the wall-clock search time in the evaluation.

The hyperparameters of the proposed method, KLENT, were unified across all five environments. Specifically, the regularization coefficients were set to $(\alpha, \beta) = (0.03, 0.1)$. The hyperparameter for $\lambda$-return was set to $\lambda = e^{-1/8} \approx 0.88$. This configuration corresponds to a time constant of 8 for the exponential decay in $\lambda$-return, indicating that returns around 8 steps ahead were considered on average. The sensitivity analysis of these hyperparameters are provided in Appendix D.

The average performances across the five environments are presented in Figure 1. The results show that the proposed method KLENT achieves the most efficient learning on average. In particular, the results indicate several-fold efficiency gains. For example, Gumbel AlphaZero required 300 million simulator evaluations to reach an average win rate of 50%, whereas KLENT required only 75 million, representing a fourfold efficiency gain.

The detailed performances in each environment are also presented in Figure 5. In Animal Shogi and Gardner Chess, where search-based approaches demonstrate high performance with moderate number of simulator evaluations, KLENT achieves competitive efficiency. In 9x9 Go, Hex, and Othello, where search-based approaches require substantial training resources, KLENT demonstrates significantly higher efficiency. In Appendix H, even in the comparison with an equal amount of test-time search, KLENT also shows a higher win rate compared to state-of-the-art Gumbel AlphaZero. In summary, our experimental results show that KLENT achieves higher win rates than search-based approaches under the same test-time computational budget, for both policies with and without test-time search.

## 5.2 ABLATION STUDY

We also conducted an ablation study to validate the importance of the three key techniques in KLENT: KL regularization, entropy regularization, and the use of $\lambda$-returns. We compared KLENT with the following four variants to evaluate the contribution of each technique. **KL Only**: Entropy regularization is removed by setting $\alpha = 0$. **ENT Only**: KL regularization is removed by setting $\beta = 0$. **1-Step KLENT**: $\lambda$-returns are replaced with 1-step backups by setting $\lambda = 0$. **Monte Carlo**

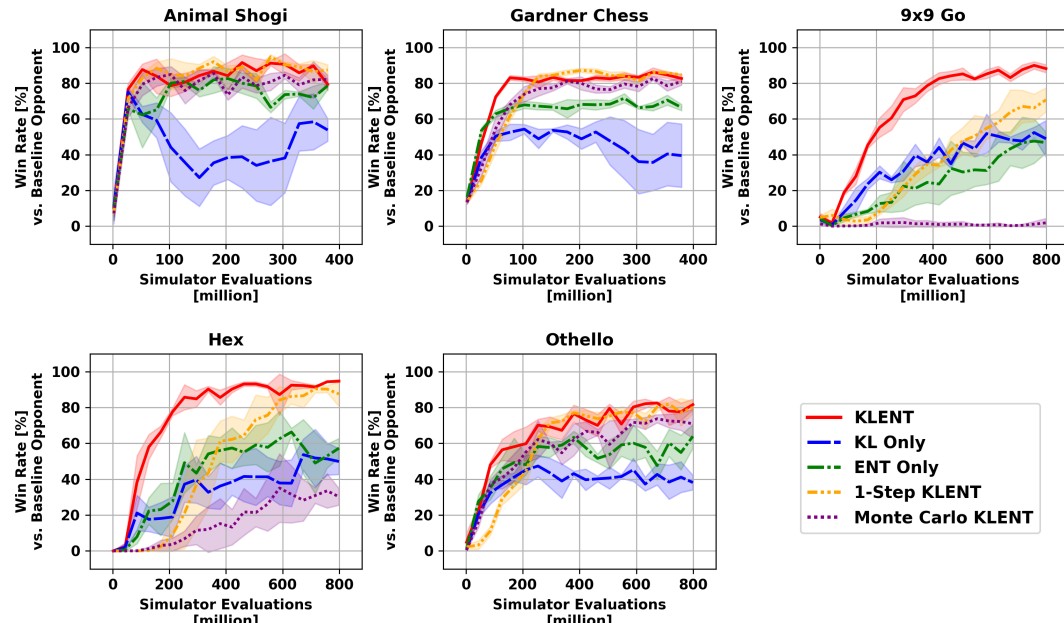

Figure 6: The results of the ablation study. They highlight the importance of all three techniques for consistently achieving high efficiency in the five environments.

**KLENT**: $\lambda$-returns are replaced with Monte Carlo returns by setting $\lambda = 1$. The hyper-parameter configurations of each variant are summarized in Table 2.

The results of our ablation study are shown in Figure 6. The results demonstrate the importance of all three techniques for consistently achieving high efficiency in the five environments. We discuss the effect of each technique below.

Table 2: Hyper-parameter configurations for the ablation study.

|  | $\alpha$ | $\beta$ | $\lambda$ |
|---|---|---|---|
| KLENT | 0.03 | 0.1 | $e^{-1/8}$ |
| KL Only | 0 | 0.1 | $e^{-1/8}$ |
| ENT Only | 0.03 | 0 | $e^{-1/8}$ |
| 1-Step KLENT | 0.03 | 0.1 | 0 |
| Monte Carlo KLENT | 0.03 | 0.1 | 1 |

**The effect of entropy regularization** can be analyzed by comparing KLENT and KL Only. In KL Only, where the entropy regularization is removed, performance degrades significantly across all the five games. Specifically, in Animal Shogi, the win rate initially rises to 75% but subsequently declines, suggesting unstable learning. Figure 7 shows the evolution of the average entropy of the policy $\pi'$ in Animal Shogi. While KLENT maintains the entropy, it rapidly decreases and becomes nearly zero in KL Only, indicating that the policy becomes excessively deterministic. These results suggest that encouraging sufficient exploration is crucial for stable learning process.

**The effect of KL regularization** can be observed by comparing the results of KLENT and ENT Only. In ENT Only, KL regularization is removed so that the policy $\pi'$ is represented by the following equation: $\pi'(a|s) = \frac{1}{Z(s)} \exp\left(Q_\theta(s,a)/\alpha\right)$. In other words, the output of the policy network is completely ignored, and actions are selected according to a softmax policy based solely on the action-value function. According to the results, ENT Only exhibits degraded performance compared to the original KLENT across the environments. The results suggest that it is also important to gradually update the policy for stabilizing the learning process.

**The effect of $\lambda$-returns** can be observed by comparing the results of KLENT, 1-Step KLENT, and Monte Carlo KLENT. Replacing $\lambda$-returns with 1-step returns or Monte Carlo returns results in a performance drop especially in 9x9 Go and Hex. As discussed in Section 4.2, the results suggest the importance of balancing bias-variance trade-off through the use of an intermediate $\lambda$.

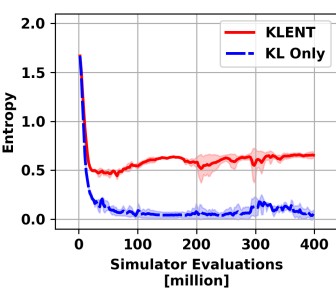 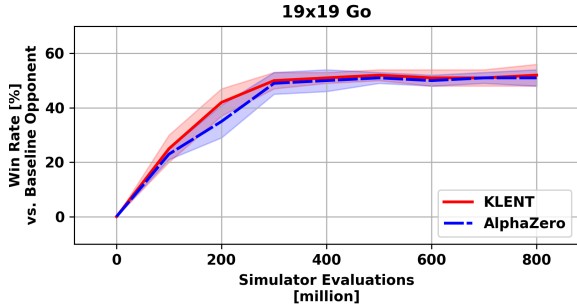

Figure 7: Entropy evolution of the policy $\pi'$ in Animal Shogi. While KLENT maintains the entropy, it becomes nearly zero in KL Only.

Figure 8: The results of experiments in 19x19 Go. Even in the large-scale environment, KLENT achieves competitive learning efficiency compared to AlphaZero.

## 5.3 Scalability to a Large-Scale Game

To assess the scalability of KLENT to a large-scale game, we further conducted experiments in 19x19 Go, comparing it with AlphaZero. As Pgx (Koyamada et al., 2023) does not provide the pretrained checkpoint for 19x19 Go, we instead used the checkpoint released by ElfOpen Go (Tian et al., 2019) for the anchored opponent. For the network architecture, we used 20-block ResNet (He et al., 2016) instead of 6-block one to capture features in the larger board. Also in this experiment, KLENT used the consistent hyperparameters, namely, $(\alpha, \beta, \lambda) = (0.03, 0.1, e^{-1/8})$.

We present the results in Figure 8. We can observe that even in 19x19 Go, KLENT achieves competitive learning compared to AlphaZero. Overall, our experimental results demonstrate that the proposed method achieves high learning efficiency in medium-scale environments, while also maintaining competitive learning in the large-scale environment.

## 6 Conclusions

In this study, we have proposed KLENT, a model-free reinforcement learning algorithm designed for board games. The key techniques used in KLENT are KL regularization for gradual policy updates, entropy regularization for exploration, and $\lambda$-returns for efficient and stable value function learning. Our experimental results have demonstrated learning efficiency of KLENT compared to existing methods. Through our ablation study, we also validated the importance of these three key techniques.

A limitation of this study is that our goal is focused to improve the efficiency. While we have shown that KLENT can achieve efficient learning, this does not necessarily mean that it achieves superior asymptotic performance when unlimited computational resources are given. Although we consider our efficient approach to be valuable for most practitioners and researchers in the community, our results do not preclude the effectiveness of search-based approaches including AlphaZero, particularly when massive computational resources such as thousands of GPUs are available.

Our results have shown that even in board games, a domain long dominated by search-based methods, carefully designed model-free approaches can achieve more efficient learning. We hope this perspective will inspire future research to extend model-free approaches to other domains where search-based methods prevail, thereby opening promising directions for resource-efficient reinforcement learning.

## Reproducibility Statement

To facilitate reproducibility, Appendix A presents the theoretical proofs, Appendix B details the experimental setup, and Appendix C details the implementation. Our Supplementary Material includes the code necessary to reproduce our experiments.

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

APPENDIX

## A    DERIVATION OF THE ANALYTICAL SOLUTION FOR THE REGULARIZED POLICY OPTIMIZATION PROBLEM

Here, we provide the formal mathematical definitions of the terms in Definition A.1 and present the proof for the derivation of the optimal solution in Equation 3 in Theorem A.2. For simplicity, we do not explicitly write the considered state $s$ in the following equations.

**Definition A.1** (KL divergence and entropy). Let $\mathcal{A}$ be a finite set and $\Delta$ be the set of all probability mass functions over $\mathcal{A}$. The Kullback-Leibler (KL) divergence between two probability mass functions $\pi' \in \Delta$ and $\pi \in \Delta$ over a finite set $\mathcal{A}$ is defined as:

$$D_{\mathrm{KL}}(\pi'\|\pi) = \sum_{a \in \mathcal{A}} \pi'(a) \log \frac{\pi'(a)}{\pi(a)}, \tag{8}$$

where it is assumed that $\pi'(a) = 0 \implies \pi'(a) \log \frac{\pi'(a)}{\pi(a)} = 0$ and $\pi(a) > 0$ for all $a \in \mathcal{A}$. The entropy of a probability mass function $\pi' \in \Delta$ over $\mathcal{A}$ is defined as:

$$H(\pi') = -\sum_{a \in \mathcal{A}} \pi'(a) \log \pi'(a), \tag{9}$$

where it is assumed that $\pi'(a) = 0 \implies \pi'(a) \log \pi'(a) = 0$.

**Theorem A.2** (Formal Derivation of the Analytical Solution $\pi'$). *Let $\mathcal{A}$ be a finite set, $\pi(a)$ a probability mass function over $\mathcal{A}$, $Q(a) : \mathcal{A} \to \mathbb{R}$ a function, and $\Delta$ the set of all probability mass functions over $\mathcal{A}$. Consider the following optimization problem:*

$$\operatorname*{maximize}_{\pi' \in \Delta} \mathbb{E}_{A \sim \pi'}[Q(A)] - \beta D_{KL}(\pi'\|\pi) + \alpha H(\pi'), \tag{10}$$

*where $\beta > 0$ and $\alpha > 0$. Then, the optimal solution is given by:*

$$\pi'(a) = \frac{1}{Z} \exp\left(\frac{Q(a) + \beta \log \pi(a)}{\alpha + \beta}\right), \tag{11}$$

*where*

$$Z = \sum_{a \in \mathcal{A}} \exp\left(\frac{Q(a) + \beta \log \pi(a)}{\alpha + \beta}\right) \tag{12}$$

*is the normalization constant.*

*Proof.* Define the Lagrangian as follows:

$$\mathcal{L}(\pi', \lambda) = \sum_{a \in \mathcal{A}} \pi'(a)Q(a) - \beta D_{\mathrm{KL}}(\pi'\|\pi) + \alpha H(\pi') - \lambda \left(\sum_{a \in \mathcal{A}} \pi'(a) - 1\right), \tag{13}$$

where $\lambda$ is the Lagrange multiplier enforcing the constraint that $\pi'(a)$ is a probability mass function.

Using the method of Lagrange multipliers, we find $\pi'$ that satisfies

$$\nabla_{\pi'}\mathcal{L}(\pi', \lambda) = 0. \tag{14}$$

Expanding this condition yields:

$$\nabla_{\pi'}\mathcal{L}(\pi', \lambda) = 0 \tag{15}$$

$$\iff \nabla_{\pi'}\left(\sum_{a \in \mathcal{A}} \pi'(a)Q(a) - \beta D_{\mathrm{KL}}(\pi'\|\pi) + \alpha H(\pi') - \lambda \left(\sum_{a \in \mathcal{A}} \pi'(a) - 1\right)\right) = 0 \tag{16}$$

$$\iff Q(a) + \beta \log \pi(a) - (\beta + \alpha)(\log \pi'(a) + 1) - \lambda = 0, \quad \forall a \in \mathcal{A} \tag{17}$$

$$\iff \log \pi'(a) = \frac{Q(a) + \beta \log \pi(a)}{\beta + \alpha} + (\text{const.}), \quad \forall a \in \mathcal{A}. \tag{18}$$

Since $\pi'(a)$ must be a probability mass function, the solution is given by Equation 11. $\square$

Here, the Lagrange multiplier $\lambda$ is unrelated to the $\lambda$ in $\lambda$-returns.

## B    Experimental Setup Details

We explain the detailed experimental setup in this section. For performance evaluation, we used the baseline opponent provided by Pgx as an anchored opponent. This anchored opponent selects actions stochastically based on its policy. The evaluated methods used deterministic policies by setting the temperature parameter to zero for softmax policies and $\epsilon$ to zero for $\epsilon$-greedy policies. In particular, the proposed method uses the greedy policy corresponding to the output $\pi$ of the policy network. In the evaluation, all agents select actions without search unifying their test-time computational resources[1]. The evaluation was conducted by playing 1024 matches against the anchored opponent, and the win rate was plotted on the vertical axis of the graph. Draws were treated as half-wins. The horizontal axis represents the total number of simulator evaluations during training, which includes all interactions with the environment simulator such as rollouts in tree search. This choice is consistent with prior literature that measures computational cost in terms of simulator evaluations, as seen in studies such as KataGo (Wu et al., 2020). Methods closer to the upper-left in the graph are interpreted as more efficient, achieving higher performance with fewer simulator accesses. For each method, experiments were conducted using three random seeds, and the mean and standard deviation of the obtained metrics were displayed on the graph.

## C    Implementation Details

This section describes the implementation details used in the experiments. For model-based methods, AlphaZero, TRPO AlphaZero, and Gumbel AlphaZero, we used open-source implementations provided by Mctx (Danihelka et al., 2022) and Pgx (Koyamada et al., 2023). Each iteration performed self-play in parallel across 1024 threads, with each thread executing up to 256 state transitions. If a game ended before 256 steps, a new game state was immediately initialized to continue the threads. Monte Carlo tree search was conducted for decision-making with a simulation budget of 32 for each action selection.

For model-free methods, including PPO, DQN, and the proposed method KLENT, self-play was similarly conducted in parallel across 1024 threads, but with each thread executing up to 2048 state transitions without search. The process for initializing new games upon completion was the same as for model-based methods. The hyperparameters of the proposed method KLENT were set as $(\alpha, \beta, \lambda) = (0.03, 0.1, e^{-1/8})$, as specified in Appendix B. The hyperparameters for PPO and DQN were determined referring to the implementation in Stable-Baselines3 (Raffin et al., 2021). For PPO, the regularization applied a clipping method to impose proximity, with the clipping ratio set to 0.2. The Generalized Advantage Estimator (GAE) in PPO used the same $\lambda = e^{-1/8}$ as KLENT. In the case of DQN, the $\epsilon$-greedy policy started with an $\epsilon$ value of 1.0, which was linearly reduced to 0.05 over the first $10^8$ simulator evaluations, and fixed at 0.05 thereafter.

The network architecture was consistent across all methods and based on ResNetV2 (He et al., 2016). The number of hidden layer channels was set to 128, for 6 residual blocks. Policy, state-value, and action-value heads were added as required by each method. Table 3 summarizes the inclusion of these heads for each method. The network takes a state observation as an input, with the policy head and action-value head outputting $|\mathcal{A}|$-dimensional vectors, and the state-value head outputting a scalar value. Due to variations in input and output shapes depending on the games and methods, the number of parameters varied slightly but remained within the range of 1.7 to 2.1 million across all experimental settings. Training of the networks was conducted with a batch size of 4096, a learning rate of 0.001, and the Adam optimizer (Kingma & Ba, 2015).

## D    Sensitivity Analysis of Hyperparameters in KLENT

This section examines the performance variation of KLENT with respect to changes in the hyperparameters $\alpha, \beta, \lambda$. Specifically, for 9x9 Go, we conducted experiments on 27 combinations of hyperparameter values as follows: $(\alpha, \beta, \lambda) \in \{0.01, 0.03, 0.1\} \times \{0.03, 0.1, 0.3\} \times \{e^{-1/4}, e^{-1/8}, e^{-1/16}\}$. For each combination, we used three random seeds and calculated the average win rate against the anchored opponent during the training steps between 600 and 800 million

---

[1]For search-based evaluations, please refer to Appendix H and I.

Table 3: Summary of the network heads included for each method.

|  | Policy Head | State-Value Head | Action-Value Head |
|---|---|---|---|
| KLENT | Yes | No | Yes |
| AlphaZero | Yes | Yes | No |
| TRPO AlphaZero | Yes | Yes | No |
| Gumbel AlphaZero | Yes | Yes | No |
| DQN | No | No | Yes |
| PPO | Yes | Yes | No |

simulator evaluations. The results are shown in Figure 9. Other experimental settings follow those described in Section 5.

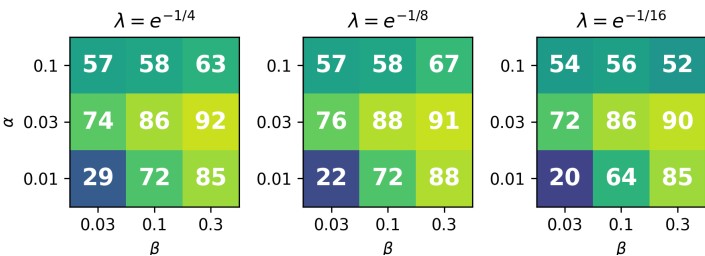

Figure 9: The results of sensitivity analysis in 9x9 Go.

When the coefficients of KL regularization and entropy regularization were both set to small values, specifically $(\alpha, \beta) = (0.01, 0.03)$, a notable decline in performance was observed. This is likely due to the improved policy, defined by Equation 3, becoming overly sharp. These results suggest that the regularization coefficients need to be set to sufficiently large and appropriate values. On the other hand, within the range of experiments conducted, the performance appears to be robust to variations in the time constant of $\lambda$-returns.

## E    DETAILS OF PRELIMINARY EXPERIMENTS ON BIAS-VARIANCE TRADE-OFF

In Figure 4, we have demonstrated the bias-variance trade-off of $\lambda$-returns in 9x9 Go environment. For the detailed experimental setup, we fixed both the policy and the value function using a pre-trained baseline model from Pgx library (Koyamada et al., 2023) in order to isolate the effect of varying $\lambda$. Since estimating the bias requires access to the true action value, which is not directly observable, we approximated the ground-truth value by computing the Monte Carlo return 1,000 times from the same state action pair and taking the average as a surrogate for the true value.

## F    ADDITIONAL EVALUATION ON THE RELIABILITY OF THE ALPHAZERO IMPLEMENTATION

### F.1    RELIABILITY OF THE PGX IMPLEMENTATION AS A BASELINE

In this study, we adopt the Pgx implementation as the baseline for AlphaZero-family methods. The original AlphaZero implementation by its authors is not publicly available. Similarly, for Gumbel AlphaZero, only the MCTS technique has been released through the Mctx library, and the full training pipeline is not open-sourced. Therefore, reproducing the full experimental setup of AlphaZero-family methods requires either relying on third-party open-source implementations or building one from scratch. To the best of our knowledge, Pgx is the only open-source implementation that satisfies all of the following criteria:

- **Peer-reviewed implementation**: Pgx was accepted to the NeurIPS 2023 benchmark track, indicating that its experimental setup has undergone peer review.

- **Evaluated across multiple environments**: Pgx has been tested on five different board games, not just a single domain. This suggests that the implementation is robust and not reliant on environment-specific tricks.

- **Performance comparison against other agents**: According to the Pgx paper, its baseline agent outperforms `pachi`, a reasonably strong Go engine.

- **Use of the `Mctx` library for MCTS**: Pgx utilizes the `Mctx` library for its MCTS technique, ensuring consistency with the Gumbel AlphaZero implementation, which was developed by some of the original AlphaZero authors.

For these reasons, we consider Pgx to be a reliable and robust open-source implementation of AlphaZero-family methods, and adopt it as the baseline in our experiments.

## F.2 PERFORMANCE COMPARISON WITH OTHER IMPLEMENTATIONS

To strengthen the credibility of the AlphaZero and baseline implementations used in this study, we conducted a comparative evaluation against a well-known open-source implementation available at https://github.com/suragnair/alpha-zero-general. This repository provides pretrained models for several games, including $8 \times 8$ Othello. We used the provided checkpoint file `pretrained_models/othello/8x8_100checkpoints_best.pth.tar` to construct an evaluation agent. We conducted a round-robin tournament involving the following four agents, where each pair played 100 games. Draws were counted as 0.5 wins for each agent.

- **Random**: An agent that selects legal moves uniformly at random.

- **AlphaZero-General**: An agent that follows the policy from the above checkpoint of `alpha-zero-general`.

- **Pgx Baseline**: The baseline agent used throughout our experiments.

- **Pgx's AlphaZero**: Our implementation of AlphaZero using the Pgx framework, trained with 800 million simulator evaluations.

The number of wins for each agent against the others is shown in Table 4. Each cell indicates the number of wins achieved by the row agent when playing against the column agent. As shown in

Table 4: Win rates among AlphaZero implementations and baselines in Othello.

|  | Random | AlphaZero-General | Pgx Baseline | Pgx's AlphaZero |
|---|---|---|---|---|
| Random | – | 3 | 0 | 3 |
| AlphaZero-General | 97 | – | 17 | 13 |
| Pgx Baseline | 100 | 83 | – | 42 |
| Pgx's AlphaZero | 97 | 87 | 58 | – |

the table, AlphaZero-General achieves a 97% win rate against the random agent, confirming that it is significantly stronger than random. However, both the Pgx Baseline and Pgx's AlphaZero implementation clearly outperform AlphaZero-General, achieving win rates of 83% and 87% respectively. These results support the reliability and strength of the implementations used in our experiments.

## G EXTENDED EXPERIMENTS ON ROLLOUT COUNTS AND TRAINING BUDGETS FOR ALPHAZERO

This section presents additional experiments to examine how AlphaZero's performance is affected by the number of rollouts per move and the total training budget.

## G.1 PERFORMANCE OF ALPHAZERO WITH VARYING ROLLOUT COUNTS IN 9X9 GO

AlphaZero performs Monte Carlo Tree Search (MCTS) at each move, where the number of rollouts corresponds to the number of simulator evaluations used per search. We investigated how this parameter affects learning efficiency.

The experiments were conducted in the 9x9 Go environment, using rollout counts of 2, 4, 8, 16, 32, and 64. The total number of simulator evaluations used during training was fixed at 200M, 400M, 600M, and 800M. Evaluation was performed by measuring the win rate against a fixed baseline agent. Note that for a fixed training budget, increasing the rollout count reduces the number of parameter updates, since each update consumes a number of simulator evaluations proportional to the rollout count. This highlights a trade-off: deeper search per move comes at the cost of fewer parameter updates. The results are shown in Table 5.

Table 5: Performance of AlphaZero with different rollout counts (9x9 Go). Each entry shows the win rate (%) against the baseline agent.

| Simulator Evaluations | 200M | 400M | 600M | 800M |
|---|---|---|---|---|
| AZ (2 rollouts) | 7 | 7 | 7 | 8 |
| AZ (4 rollouts) | 16 | 35 | 51 | 61 |
| AZ (8 rollouts) | *20* | *39* | *56* | *69* |
| AZ (16 rollouts) | 15 | 28 | 42 | 57 |
| AZ (32 rollouts) | 6 | 13 | 20 | 34 |
| AZ (64 rollouts) | 5 | 7 | 11 | 15 |
| (cf: KLENT) | **53** | **80** | **85** | **89** |

The results indicate that in 9x9 Go, setting the rollout count to around 8 leads to the most efficient learning for AlphaZero. Nevertheless, even when the rollout count is optimized, KLENT achieves substantially higher performance under the same training budget, highlighting its superior efficiency.

## G.2 PERFORMANCE OF ALPHAZERO WITH VARYING ROLLOUT COUNTS IN 19X19 GO

We also tuned the number of rollouts in 19×19 Go with values of 4, 16, 64, and 256. As shown in Figure 10, 16 rollouts achieved the most efficient learning. Accordingly, we reported this result as the performance of AlphaZero in Figure 8 in Section 5.3.

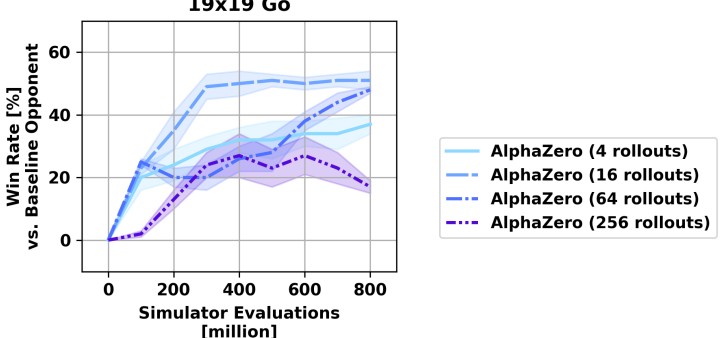

Figure 10: The results of rollout count tuning in 19x19 Go. 16 rollouts achieve the most efficient learning.

## G.3 PERFORMANCE OF ALPHAZERO WITH INCREASED TRAINING BUDGETS

We also conducted additional experiments to examine AlphaZero's asymptotic performance by increasing the total training budget. The experimental settings were the same as above, and the number of simulator evaluations was extended up to 4,800M. The results are presented in Table 6.

Table 6: Performance of AlphaZero under increased training budgets (9x9 Go). Each entry shows the win rate (%) against the baseline agent.

| Simulator Evaluations | 800M | 1,600M | 2,400M | 3,200M | 4,000M | 4,800M |
|---|---|---|---|---|---|---|
| AZ (2 rollouts) | 8 | 18 | 17 | 18 | 19 | 18 |
| AZ (4 rollouts) | 61 | 75 | 75 | 85 | 86 | 85 |
| AZ (8 rollouts) | 69 | 79 | 85 | 85 | 85 | 86 |
| AZ (16 rollouts) | 57 | 80 | 83 | 88 | **89** | **89** |
| AZ (32 rollouts) | 34 | 60 | 71 | 78 | 83 | 83 |
| AZ (64 rollouts) | 15 | 34 | 51 | 59 | 67 | 72 |
| (cf: KLENT) | **89** | – | – | – | – | – |

These results show that AlphaZero reaches approximately 89% win rate when the total training budget is increased to around 3,200M to 4,000M simulator evaluations. This confirms the intuitive expectation that AlphaZero can achieve strong asymptotic performance given sufficient training budget. At the same time, KLENT achieves comparable performance using only 800 million simulator evaluations, which is approximately four to five times fewer than those required by AlphaZero, underscoring its efficiency advantage.

## H  STRENGTH SCALING WITH ADDITIONAL TEST-TIME COMPUTATION

Additional simulations during test time can improve the strength of agents. In this section, we investigate how performance scales with the number of simulations for models trained with KLENT and those trained with Gumbel AlphaZero in 9x9 Go. For both methods, parameters trained with 800 million simulator evaluations are used. We adopt an off-the-shelf Gumbel AlphaZero Monte Carlo Tree Search (MCTS) for test-time computation, applying the same procedure to both sets of parameters. While Gumbel AlphaZero learns policy and state-value networks, KLENT trains policy and action-value networks. To address this difference, for KLENT, the inner product of the policy and action-value is used as the state-value estimate during MCTS. The anchored baseline opponent uses parameters provided by Pgx and runs with 800 simulations. Koyamada et al. (2023) have reported that this agent has achieved 62 wins and 38 losses against Pachi (Baudiš & Gailly, 2011) with 10,000 simulations. We measure the win rates of the evaluated target agents, using either KLENT or Gumbel AlphaZero parameters, under 0, 16, 32, 64, 100, 200, 400, and 800 simulations. Here, 0 indicates that the agent conducts no search and deterministically chooses action solely based on its policy network. In this experiment, the evaluation is conducted for 100 matches. The win rates are measured with three random seeds and the mean and the standard deviation are plotted.

The results are shown in Figure 11, where the horizontal axis represents the number of simulations and the vertical axis represents win rates against the anchored baseline. KLENT demonstrates that it can effectively scale its strength with test-time computation. In this experiment, we calculate the state-values from policy and action-value estimates as

$$V^\pi(s) = \sum_a \pi(a|s)Q^\pi(s,a),$$

which is based on the policy $\pi$ we actually use for rollout.

**Wall-clock Inference Time:**  An evaluation with equal wall-clock search time is a fair condition for performance comparison. Actually, as we use the same ResNet architecture for each method, the wall-clock search time is proportional to the test-time rollout count. Therefore, the equal wall-clock search time comparison is equivalent to our equal rollout count comparison. To further verify this point, we have measured the wall-clock time spent for each action selection and summarized in the following table.

The results show there is no significant difference between wall-clock inference time of Gumbel AlphaZero and KLENT + test-time MCTS. Therefore, we conclude that the evaluation with fixed rollout counts is equivalent to equal wall-clock time evaluation.

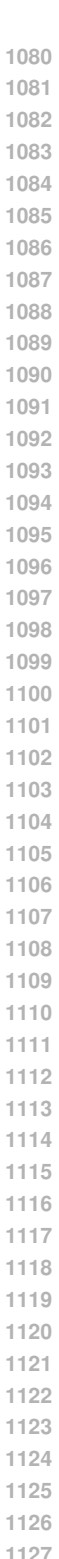

Figure 11: Performance changes with increased test-time computation budget. The simulation budget of the anchored baseline opponent is fixed at 800. The horizontal axis represents the simulation budget for the evaluated agents, while the vertical axis shows their win rate against the anchored opponent. The results demonstrate that agents using parameters trained with KLENT can scale their strength as the number of test-time simulations increases.

Table 7: Test-time rollout counts and wall-clock inference time.

| Rollout Count | Gumbel AlphaZero | KLENT + Test-Time MCTS |
|---|---|---|
| 0 | $(7.128 \pm 0.400) \times 10^{-4}$ | $(7.157 \pm 0.084) \times 10^{-4}$ |
| 16 | $(2.601 \pm 0.013) \times 10^{-2}$ | $(2.603 \pm 0.012) \times 10^{-2}$ |
| 32 | $(5.403 \pm 0.019) \times 10^{-2}$ | $(5.359 \pm 0.026) \times 10^{-2}$ |
| 64 | $(1.093 \pm 0.006) \times 10^{-1}$ | $(1.100 \pm 0.005) \times 10^{-1}$ |
| 100 | $(1.732 \pm 0.007) \times 10^{-1}$ | $(1.734 \pm 0.007) \times 10^{-1}$ |
| 200 | $(3.435 \pm 0.010) \times 10^{-1}$ | $(3.444 \pm 0.017) \times 10^{-1}$ |
| 400 | $(6.918 \pm 0.041) \times 10^{-1}$ | $(6.970 \pm 0.021) \times 10^{-1}$ |
| 800 | $(1.391 \pm 0.009) \times 10^{0}$ | $(1.389 \pm 0.004) \times 10^{0}$ |

## I HEAD-TO-HEAD MATCHES

### I.1 EVALUATION AGAINST PACHI AND GNUGO IN 9x9 GO

In the domain of 9x9 Go, we conducted additional head-to-head experiments against GnuGo and Pachi, which are baselines confirmed to have been used in prior studies. The detailed configurations of these agents are provided below.

- Evaluated Agent
  - KLENT: The model trained with KLENT. Similarly to Appendix H, Gumbel AlphaZero was employed as the search algorithm at test time, with the number of rollouts set to 2,000 (approximately two seconds per move). For the neural network parameters, we used the model trained by KLENT with 800M simulator evaluations. While the MCTS in Gumbel AlphaZero requires estimates of the policy and state value, KLENT's neural network estimates the policy and action values. To account for this difference, we used the inner product of the policy and action-value predictions as the state-value estimate.

- Anchored Opponent
  - GnuGo (Bump et al., 2005): A classical and lightweight MCTS-based Go engine. The strength level was set to 10 (the strongest level), following the evaluation setting in prior work (Hessel et al., 2021).
  - Pachi (Baudiš & Gailly, 2011): A fairly strong MCTS-based Go engine. This program has been reported to have the strength of a KGS 7-dan player in 9x9 Go (Baudiš & Gailly, 2018), which corresponds to the top 0.5–1% of players on Kiseido Go Server. The strength was set by configuring the MCTS rollout count to 10,000, consistent

| Anchored Opponent | Winrate of KLENT's side |
|---|---|
| GnuGo (Level 10) | 100% |
| Pachi (10K rollouts) | 81% |

Table 8: The results of head-to-head matches against GnuGo and Pachi.

with the evaluation settings in prior work (Hessel et al., 2021; Danihelka et al., 2022; Koyamada et al., 2023).

Under these conditions, we conducted 100 games, and the win rate of KLENT is presented in Table 8. These results demonstrate the win rates against agents that have been used for evaluation in prior studies, and we believe they can serve as one of the credible reference points.

### I.2 HEAD-TO-HEAD MATCH AGAINST ALPHAZERO IN 19x19 GO

We additionally conducted direct head-to-head matches between the final checkpoints trained in Section 5.3. In this setting, the evaluation used MCTS with 800 rollouts per move. The AlphaZero checkpoint was trained with 16 rollouts, which was the strongest among the tested settings. Under this protocol, KLENT won all evaluation games, yielding a 100% win rate against AlphaZero trained with the same simulator budget. These result also support that KLENT can achieve efficient learning under a fixed training budget.

## J  PERFORMANCE COMPARISON IN ELO RATINGS

While win rate was used as the primary metric for comparing trained agents in the main paper, for reference, we provide Elo scores in Figure 12. Specifically, we fix the Elo score of the Pgx baseline agent at $R_0 = 1000$, and apply the following standard formula for Elo rating:

$$R = 400 \log_{10}\left(\frac{W}{L}\right) + R_0,$$

where $W$ denotes the win rate against the Pgx baseline and $L = 1 - W$ is the corresponding loss rate. Since the mapping from win rate to Elo is monotonic, this transformation does not alter our primary claim that KLENT outperforms the baselines under a fixed computational budget. However, Elo scores must be interpreted with care, as they are highly sensitive to the composition of the tournament pool. Indeed, in our preliminary experiments, we observed that Elo ratings of fixed agents could vary significantly when the set of evaluated agents is modified. This sensitivity has also been pointed out in prior works (Balduzzi et al., 2018; Liu et al., 2025; Lanctot et al., 2025). These studies highlight that Elo ratings can be manipulated by adding redundant or biased agents, even when anchor points are fixed. Therefore, cross-paper comparisons of Elo scores require identical tournament configurations, which is difficult in our case since neither the full tournament details of the Pgx implementation nor those of Gumbel AlphaZero are publicly available. For this reason, we present Elo scores only as supplementary information.

## K  COMPUTATIONAL REQUIREMENTS

For overall experiments, we have spent approximately 2,000 GPU hours on NVIDIA A100 GPU in total to run the main experiments in Figure 5 to 8. KLENT algorithm can be run on a single NVIDIA A100 GPU. This section describes the computational and memory requirements of the algorithm.

**Memory Usage**   KLENT stores improved policies in a replay buffer for reuse. In our experiments, memory usage was not an issue on a single A100 GPU with 80 GB of memory. Even when memory becomes a limiting factor, this issue can be mitigated using a sparse representation. Since the improved policy assigns non-zero probabilities only to legal actions and sets all others to zero, sparse storage formats can significantly reduce memory consumption.

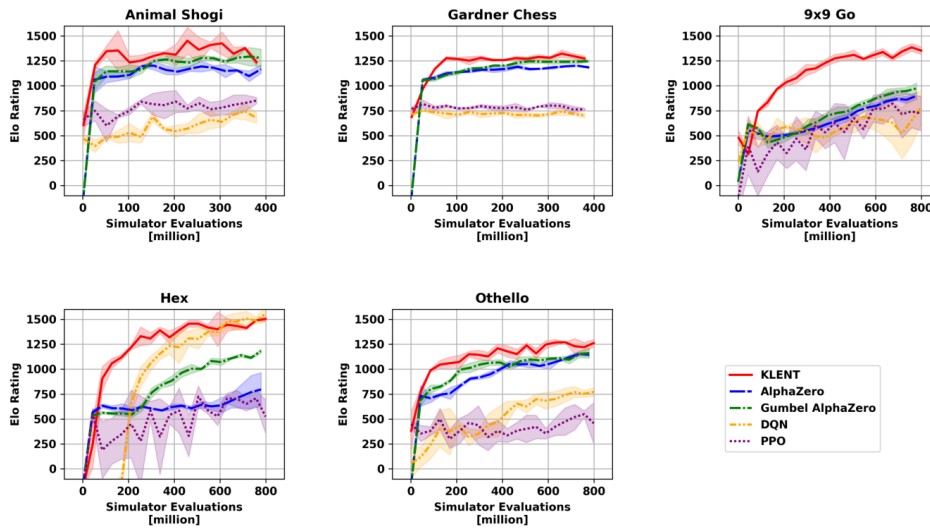

Figure 12: **Performance comparison in Elo scores.** Win rates are converted by fixing the Pgx baseline to Elo 1000. Note that Elo-based cross-paper comparisons are unreliable due to sensitivity to tournament configurations.

To illustrate this, we collected states from 10,000 games played by baseline agents implemented with Pgx and computed the average and maximum number of legal actions per game. The results are shown in the table below.

Table 9: Statistics of legal actions collected from 10,000 games for each environment.

| Game | Action Space Size | Mean Legal Actions | Max Legal Actions |
|---|---|---|---|
| Animal Shogi | 132 | 7.5 | 36 |
| Gardner Chess | 1,225 | 9.5 | 40 |
| 9x9 Go | 82 | 42.3 | 82 |
| Hex | 122 | 90.6 | 121 |
| Othello | 65 | 8.0 | 22 |

These results indicate that the number of legal actions is often much smaller than the full action space. Therefore, sparse representations provide an effective solution in memory-constrained settings.

**Computation Time** One of KLENT's strengths lies in its training efficiency. For example, in the 9x9 Go environment, KLENT reduced the time required to surpass the baseline agent by more than 25% compared to Gumbel AlphaZero and AlphaZero.

This efficiency stems from KLENT requiring fewer simulator interactions and neural network evaluations per training sample. As a result, it offers practical advantages in terms of wall-clock training time and computational cost.

## L    CONVERGENCE LIMIT OF KLENT

In this section, we examine the convergence limit of KLENT. In two-player games, quantal response equilibrium (McKelvey & Palfrey, 1995) is defined as a policy which satisfies the following equa-

tion:

$$\pi(a|s) = \frac{1}{Z(s)} \exp(Q^\pi(s,a)/\alpha).$$

Sokota et al. (2022) have provided a theoretical analysis on the combination of KL and entropy regularization, and it suggests that the convergence limit of this combination is the quantal response equilibrium. To verify this expectation, we have conducted experiments on a simple small-scale game, namely, the count-up game. The rule is defined as follows.

**Formal Rule**  Let us consider a two-player sequential zero-sum game. Let $N$ and $k$ be positive integers. The state space is non-negative integers $\mathcal{S} = \{0, 1, 2, \dots\}$ and the action space is positive integers up to $k$: $\mathcal{A} = \{1, \dots, k\}$. The initial state $S_0$ is always 0, and the state transition, termination, and rewards are defined as follows.

- Transition: The next state is defined as $S_{t+1} = S_t + A_t$.
- Termination: The game terminates when $S_t + A_t \geq N$.
- Rewards: The reward is defined as follows:

$$r(S_t, A_t) = \begin{cases} 1 & \text{if } S_t + A_t \geq N \\ 0 & \text{otherwise} \end{cases}$$

**Interpretation of the Rule**  This rule can be interpreted as follows. There are two players and they declare the number of $S_{t+1}$ alternately. Let $N = 7$ and $k = 2$, for example. Then, the first player can declare 1 or 2, and the next player can declare a number that is larger by 1 or 2 than the previously declared number. If a player declares a number which is equal to or larger than 7, the player wins the game.

In this simple and small-scale game, we analyze the convergence limit of KLENT. Below, we assume that $N = 7$ and $k = 2$ unless otherwise described.

**Optimal Strategy**  The optimal strategy of this game can be calculated in a backward manner as Table 10.

| State $S_t$ | Optimal Strategy |
| :---: | :---: |
| 6 | Win with $A_t = +1$ or $+2$. |
| 5 | Win with $A_t = +2$. |
| 4 | Lose anyway. |
| 3 | Win with $A_t = +1$. |
| 2 | Win with $A_t = +2$. |
| 1 | Lose anyway. |
| 0 | Win with $A_t = +1$. |

Table 10: Optimal strategy in the count up game.

**Quantal Response Equilibrium**  In this game, quantal response equilibrium can also be calculated in a backward manner. We have calculated them for $\alpha = 0.03$ and $\alpha = 1.0$. The results are shown in Figure 13. It can be observed that the equilibrium of $\alpha = 0.03$ is close to the optimal strategy in Table 10, and that of $\alpha = 1.0$ is a softer policy.

**Convergence Limit of KLENT**  We have run the KLENT algorithm on this game, especially with $\alpha = 1.0$. The evolution of learned policy and action values is shown in Figure 14. The results confirm the expectation that KLENT converges to the quantal response equilibrium.

## M  THE USE OF LARGE LANGUAGE MODELS

We have utilized large language models to polish our writing and correct grammatical errors.

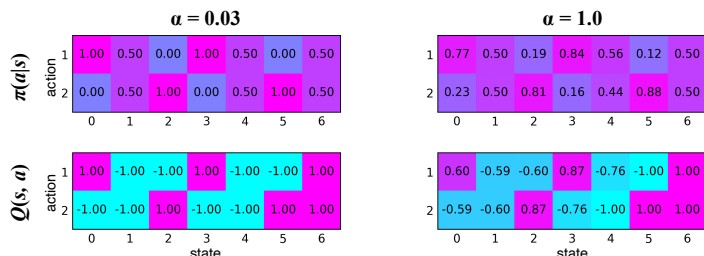

Figure 13: Quantal response equilibrium in the count-up game.

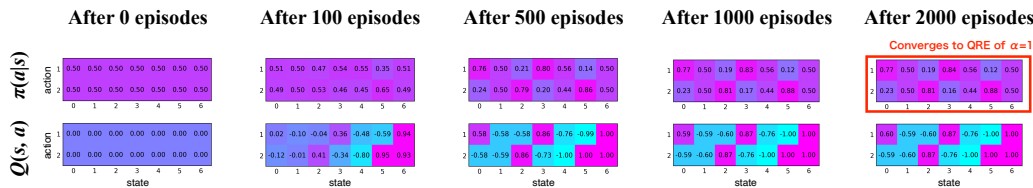

Figure 14: The evolution of learned policy and action values of KLENT in the count-up game. The results show that KLENT converges to the quantal response equilibrium.