# OpenReview forum: "Resource-Efficient Model-Free Reinforcement Learning for Board Games"
_ICLR.cc/2026/Conference — Submitted to ICLR 2026_

### Official Review · Reviewer_iDUE · 2025-10-20

**Soundness:** 2
**Presentation:** 3
**Contribution:** 2
**Rating:** 4
**Confidence:** 4

**Summary:**

This paper proposes the KLENT algorithm for board games to achieve more efficient learning. Both KL regularization and entropy regularization are used to assist the training of the policy network, while λ-returns are employed to train the value function network. Experiments are conducted on five board games: Animal Shogi, Gardner Chess, Go, Hex, and Othello, and the ablation study demonstrates the contribution of each component.

**Strengths:**

1. The paper is well-organized and easy to follow.

2. The paper includes extensive experiments conducted on five game environments, along with numerous ablation studies.

**Weaknesses:**

1. The comparison with baseline algorithms in the experiments is not very clear, so I cannot be sure whether the proposed KLENT algorithm actually outperforms AlphaZero. For more details, please refer to the Question section.

2. The paper’s originality is somewhat limited: using $\lambda$-returns to train the value function network is a common practice, and employing the exact solution of the optimization objective to train the policy is also used before. Therefore, it seems that the core innovation of KLENT lies mainly in the simultaneous use of KL and entropy regularization. As illustrated in Line 131, "we aim to empirically show that this combination is effective for achieving efficient learning in the board game domain."

**Questions:**

1. Grill et al. (2020) used KL regularization to constrain the learning of the policy and replaced MCTS with an exact solution, which is very similar to KLENT. Comparing Equation (2) in this paper with Equation (7) in Grill et al. (2020), it appears that the main difference is the additional entropy term. Could you carefully analyze the differences between the two and include Grill et al. (2020) as a comparison model in the experiments?

2. In the evaluation, is the policy learned by KLENT used to guide MCTS for comparison with AlphaZero, or is the comparison based solely on the policy learned by the neural network without MCTS?

3. Using the number of simulator evaluations as a resource constraint makes sense, and I understand that approach. However, I am also curious: in AlphaZero, collecting a single training data point via MCTS requires multiple neural network calls, whereas KLENT does not. I would like to know the respective amounts of data collected by KLENT and AlphaZero, and how much difference there is in their actual runtime.

4. The performance of DQN in Hex is comparable to KLENT, but it performs poorly in the other games. Can you explain this phenomenon?

5. How many MCTS simulations does AlphaZero perform during training and evaluation?

Grill, Jean-Bastien, et al. "Monte-Carlo tree search as regularized policy optimization." International Conference on Machine Learning. PMLR, 2020.

---

> ### Author Response · Authors · 2025-11-21
>
> We thank the reviewer for the thorough review and constructive feedback about this work. We consider that the reviewer has kindly raised weaknesses and questions on the following topics.
>
> * Weaknesses
>     * W1: **Experimental Details** (Split into Q2, Q3, and Q5.)
>     * W2: **Originality**
> * Questions
>     * Q1: **Comparison to Grill+**
>     * Q2: **Evaluation Protocol**
>     * Q3: **Data Collected and Actual Runtime**
>     * Q4: **DQN on Hex**
>     * Q5: **MCTS Simulation Count**
>
> We would like to answer the above as follows.
>
> **W2: Originality**
>
> Thank you for raising a discussion on the originality of our paper. We acknowledge that the three core techniques used in KLENT, namely KL regularization, entropy regularization, and λ-returns, are standard techniques in the RL field. Therefore, we consider our contributions to be empirical rather than algorithmic. In our extensive experiments, we have demonstrated that properly revisiting existing techniques can achieve efficient learning in the board game domain, and also shown that the proper combination is important to achieve high performance through ablation studies. We consider these empirical results to be valuable and original, as search-based methods have been believed to be state-of-the-art in this domain.
>
> In response to your review, we have revised the second-to-last paragraph of the introduction section to properly explain our contributions. We would like to thank you for your comprehensive review.
>
> **Q1: Comparison to Grill+**
>
> Thank you for raising a question on the comparison to Grill et al. (2020). In addition to the entropy term, there are differences between KLENT and Grill et al. (2020) as follows.
>
> * Use of MCTS: Grill et al. (2020) use MCTS to estimate action values $Q(s, a)$. On the other hand, KLENT directly uses the output of the action value network to obtain the optimal policy. This elimination of MCTS makes KLENT a simple model-free algorithm.
> * Forward vs. reverse KL regularization: Grill et al. (2020) consider a policy optimization problem with forward KL regularization to theoretically approximate the original AlphaZero. This optimization problem is difficult to analytically solve, so they use dichotomic search to obtain the optimal solution to it. On the other hand, KLENT considers reverse KL regularization as in Equation 2 of our paper, resulting in a simple analytical softmax solution in Equation 3 of our paper. We consider our algorithm simpler and easier to implement, as we do not need a dichotomic search to obtain the optimized policy.
>
> In response to your kind review, we have revised the last paragraph of Section 4.1 to clarify these differences. The paragraph also includes the comparison between KLENT and other prior studies, based on the comments by Reviewer 9LfE and Reviewer bWVB.
>
> For the performance of the MCTS of Grill et al. (2020) in the board game domain, the Gumbel AlphaZero paper (Danihelka et al., 2022) reports that their algorithm outperforms Grill et al. (2020). Therefore, we expect that KLENT, which achieves higher efficiency than Gumbel AlphaZero, can also achieve higher efficiency than the MCTS of Grill et al. (2020). As the reviewer has kindly suggested, we consider that the direct comparison of KLENT and Grill et al. (2020) is still important. We plan to include it as a comparison model in the experiments and add the results to the paper as soon as possible. Thank you for your kind comments.
>
> **Q2: Evaluation Protocol**
>
> Our main experiment is the comparison based solely on the policy learned by the neural network without test-time MCTS. For performance scaling of KLENT with test-time MCTS, we have provided additional results in Appendix H. The results in Figure 11 show that the network learned by KLENT can improve its strength with test-time MCTS.
>
> In response to your thoughtful review, we have added a fourth paragraph to Section 5.1 to clarify this point. Thank you for your kind review.
>
> **Q3: Data Collected and Actual Runtime**
>
> Thank you for raising a question on the number of collected data and the actual runtime comparison. For search-based methods (AlphaZero and Gumbel AlphaZero), the number of data collected is the number of simulator evaluations divided by the number of MCTS simulations, which is set to be 32 in Figure 5. For model-free methods (KLENT, DQN, and PPO), the number of data collected is equal to the number of simulator evaluations. In response to your kind review, we have revised our paper clarifying this point in the second paragraph of Section 5.1.
>
> For actual runtime, although KLENT collects more samples than search-based methods, KLENT still achieves faster learning than other approaches. For example, KLENT reduced the overall training runtime required to outperform the baseline agent by more than 25% compared to Gumbel AlphaZero and AlphaZero in 9x9 Go. Further computational requirements of KLENT are provided in Appendix K.
>
> ---
>
> *(to be continued in the next comment)*

---

> > ### Author Response · Authors · 2025-11-21
> >
> > *(continued from the previous comment)*
> >
> > **Q4: DQN on Hex**
> >
> > Thank you for reviewing our experimental results in detail. While we did not specifically analyze the behavior of DQN on Hex as it is not our main focus, we consider the results are consistent with prior studies, including NeuroHex (Young et al. 2016), which has shown that DQN can learn a somewhat strong policy in Hex.
> >
> > It is possible to consider that this phenomenon is caused by the characteristics of the game rules. In Hex, the game state gradually changes as one action can add only one stone to the board. However, in other games, one action can dramatically change the state, removing or flipping multiple stones in Go and Othello, or taking and promoting pieces in Animal Shogi and Gardner Chess, for example. Considering that DQN has demonstrated successful results in video games such as Atari, where the state transition is somewhat gradual, the gradual state transition of Hex might be preferred by DQN.
> >
> > **Q5: MCTS Simulation Count**
> >
> > AlphaZero uses 32 and 16 MCTS simulations in Figures 5 and 8, respectively. The results with other MCTS simulation counts are provided in Appendix G.
> >
> > **References**
> >
> > * Ivo Danihelka et al. Policy Improvement by Planning with Gumbel. ICLR 2022.
> > * Kenny Young et al. NeuroHex: A Deep Q-learning Hex Agent. IJCAI Computer Games Workshop 2016.
> >
> > ---
> >
> > We thank the reviewer again for the helpful feedback and please let us know if there are any additional concerns or questions.

---

> ### Author Response · Authors · 2025-11-25
>
> Thank you again for your kind review.
>
> **Q1: Comparison to Grill+ (update)**
>
> In response to your kind suggestion in the first question (Q1), we have run additional experiments for Grill et al., (2020) and updated Figures 1 and 5, third paragraph of Section 5.1, and Appendix C accordingly. As the original paper (Grill et al., 2020) does not specify the name of the proposed method, we refer to it as TRPO AlphaZero, following the manner in Gumbel AlphaZero paper (Danihelka et al., 2022). The results in Figure 5 show that it demonstrate good performance on Animal Shogi, but not so much in other environments. The average performance was close to AlphaZero as shown in Figure 1.
>
> Thank you for your helpful comments to improve our paper. If you have any further concerns or questions, we would appreciate it if you could let us know.

---

### Official Review · Reviewer_vsPq · 2025-10-25

**Soundness:** 2
**Presentation:** 3
**Contribution:** 2
**Rating:** 4
**Confidence:** 4

**Summary:**

The paper proposes KLENT, a model-free reinforcement learning algorithm for board games. The method is presented as a "simple" alternative to complex search-based methods like AlphaZero. KLENT combines three existing RL techniques: Kullback-Leibler (KL) regularization for gradual policy updates, entropy regularization for exploration, and $\lambda$-returns for value function learning. The authors conduct experiments on five board games (e.g., Animal Shogi, 9x9 Go, Hex) and argue that KLENT is significantly more "resource-efficient" in terms of training-time simulator evaluations compared to AlphaZero, Gumbel AlphaZero, DQN, and PPO. An ablation study is also presented to show that all three components are necessary for this efficiency.

**Strengths:**

The paper is well-written and easy to follow. The experiments are relatively extensive, covering five different games and including a detailed ablation study (Section 5.2) and hyperparameter analysis (Appendix D), which validate the components of the proposed method. The main result on training efficiency is interesting; for example, in 9x9 Go, KLENT achieves a high win rate (~89%) using only 800M simulator evaluations, while the paper's AlphaZero implementation requires ~4000M evaluations to reach a similar level (Appendix G.3, Table 6). This demonstrates good sample efficiency for learning a good reactive policy.

**Weaknesses:**

My main concerns with this paper are its limited novelty and the questionable fairness of its central experimental comparison.

1.  **Limited Novelty:** The paper's primary contribution is applying a combination of existing techniques to a new domain. The "KLENT" algorithm is a straightforward combination of KL regularization, entropy regularization, and $\lambda$-returns. As the authors note in Section 3.1, the combination of KL and entropy regularization has already been theoretically analyzed in prior work (e.g., Vieillard et al., 2020 and [1]). Furthermore, $\lambda$-returns are a foundational technique in RL (Sutton, 1988). Thus, the paper is an application study, not a proposal of a new algorithm.

2.  **Lack of Domain-Specific Insight:** The proposed method is a general model-free RL algorithm. It doesn't seem to incorporate any specific inductive biases or improvements for two-player, zero-sum, perfect-information games, which is the entire domain of study. This makes it less of a direct competitor to AlphaZero, which is fundamentally built on MCTS as a policy improvement operator that leverages the game structure.

3.  **Flawed Core Comparison:** The paper's central claim of superior "efficiency" (e.g., in Figure 1 and 5) is based on a flawed premise. The main experiments evaluate all agents, including AlphaZero, *without* search at test time. AlphaZero is an algorithm *designed* to use search (MCTS) to plan; evaluating its learned policy network in a purely reactive way is not a meaningful measure of its performance. This setup inherently favors the model-free KLENT, which is trained specifically to produce a strong reactive policy. The paper's metric of "simulator evaluations" also penalizes search-based methods during training by counting MCTS rollouts, which is what they are *supposed* to do.

4.  **Misleading Scalability Argument:** The paper frames KLENT as a "resource-efficient" alternative. However, the strength of AlphaZero is its ability to scale performance with *computational complexity* (i.e., more search). While Appendix H attempts to show KLENT scaling with test-time search, it relies on a heuristic (using the policy-Q inner product as a value estimate) and the performance gains are minimal. This suggests the learned model is not well-suited for deep planning. The paper's own data in Table 6 shows that AlphaZero *does* eventually match KLENT's performance, it just takes a larger training budget. This implies KLENT is *sample-efficient* at learning a good-enough policy, but it is not a more scalable or asymptotically superior approach, which is the key to superhuman performance in this domain.

[1] Lee D. Entropy-Augmented Entropy-Regularized Reinforcement Learning and a Continuous Path from Policy Gradient to Q-Learning

**Questions:**

1.  The novelty of KLENT seems to be the *empirical* combination of KL-reg, entropy-reg, and $\lambda$-returns. Given that the paper cites prior work like Vieillard et al. (2020) that already analyzes the KL/entropy combination, could you please clarify what the *algorithmic* contribution is?

2.  Your main results in Figure 5 are based on evaluating AlphaZero "without search". How can this be considered a fair comparison when AlphaZero's entire design is centered on using search to improve its policy at decision time? Would not a comparison with equal *wall-clock* search time at evaluation be more appropriate?

3.  In Appendix H, you plug the KLENT-trained network into an MCTS. This relies on a heuristic $V(s) = \sum_a \pi(a|s)Q(s,a)$. How much does this heuristic limit the search? Does the small performance gain (Fig 11) not simply confirm that KLENT, as a model-free method, is not designed to scale with computational search in the same way AlphaZero is?

4. Table 6 shows AlphaZero eventually matching KLENT's peak performance, albeit with a 5x larger training budget. Does this not frame the paper's contribution as one of *sample efficiency* to a certain performance level, rather than a truly superior or more scalable algorithm?

---

> ### Author Response · Authors · 2025-11-21
>
> We would like to thank the reviewer for carefully reading even our appendices and for the constructive feedback on this work. We consider that the reviewer has kindly raised weaknesses and questions on the following topics.
>
> * Weaknesses
>     * W1: **Novelty**
>     * W2: **Domain-Specific Knowledge**
>     * W3: **Comparison Protocol**
>     * W4-1: **KLENT with MCTS**
>     * W4-2: **Framing of Contributions**
> * Questions
>     * Q1: **Novelty**
>     * Q2: **Comparison Protocol**
>     * Q3: **KLENT with MCTS**
>     * Q4: **Framing of Contributions**
>
> We would like to answer the above as follows.
>
> **W1, Q1: Novelty**
>
> Thank you for raising a discussion on the novelty of this work. As the reviewer has kindly pointed out, our main contributions are empirical rather than algorithmic.
>
> However, we consider our empirical results bring valuable knowledge to ICLR community. We have demonstrated that properly revisiting existing techniques can achieve efficient learning in the board game domain through our experiments, and also shown that the proper combination is important to achieve high performance through ablation studies. We consider these empirical results to be original and valuable to the community, as search-based methods have been believed to be state-of-the-art in this domain.
>
> In response to your kind review, we have revised the second-to-last paragraph of the introduction section to properly explain our contributions. We would like to thank you for your comprehensive and helpful review.
>
> **W2: Domain-Specific Knowledge**
>
> Thank you for mentioning that the proposed method is a general model-free RL algorithm. We acknowledge that domain-specific knowledge, such as game tree structure, has been utilized to design MCTS-based methods such as AlphaZero. On the other hand, we have proposed a general algorithm without such domain-specific knowledge, and experimentally shown that it can achieve more sample-efficient learning than MCTS-based methods. We consider that this result suggests such domain-specific knowledge may not be necessary to efficiently learn to play board games, and that this finding is valuable and novel to the community.
>
> **W3, Q2: Comparison Protocol**
>
> Thank you for asking a question on the fairness of our evaluation protocol. As the reviewer has kindly mentioned, an evaluation with equal wall-clock search time is a fair condition for performance comparison. Actually, as we use the same ResNet architecture for each method, the wall-clock search time is proportional to the test-time rollout count. Therefore, the equal wall-clock search time comparison is equivalent to our equal rollout count comparison. To further verify this point, we have measured the wall-clock time spent for each action selection and summarized in the following table.
>
> | Test-Time Rollout Count | Gumbel AlphaZero Inference Time [sec] | KLENT + Test-Time MCTS Inference Time [sec] |
> | ----------------------- | ---------------------------------------- | ----------------------------------------------- |
> | 0                       | $(7.128 ± 0.400) × 10^{-4}$                    | $(7.157 ± 0.084) × 10^{-4}$                           |
> | 16                      | $(2.601 ± 0.013) × 10^{-2}$                    | $(2.603 ± 0.012) × 10^{-2}$                           |
> | 32                      | $(5.403 ± 0.019) × 10^{-2}$                    | $(5.359 ± 0.026) × 10^{-2}$                           |
> | 64                      | $(1.093 ± 0.006) × 10^{-1}$                    | $(1.100 ± 0.005) × 10^{-1}$                           |
> | 100                     | $(1.732 ± 0.007) × 10^{-1}$                    | $(1.734 ± 0.007) × 10^{-1}$                           |
> | 200                     | $(3.435 ± 0.010) × 10^{-1}$                    | $(3.444 ± 0.017) × 10^{-1}$                           |
> | 400                     | $(6.918 ± 0.041) × 10^{-1}$                    | $(6.970 ± 0.021) × 10^{-1}$                           |
> | 800                     | $(1.391 ± 0.009) × 10^{0}$                     | $(1.389 ± 0.004) × 10^{0}$                            |
>
> The results show there is no significant difference between wall-clock inference time of Gumbel AlphaZero and KLENT + test-time MCTS. Based on these reasons, we consider our comparison in Figures 5 and 11 is a fair evaluation, as it is equivalent to an equal wall-clock search time comparison. In response to your thoughtful review, we have added a fourth paragraph to Section 5.1 to clarify this equivalence to the readers. We have also added the results above in Table 7 of Appendix H. Thank you for raising a question on this point.
>
> ---
>
> *(to be continued in the next comment)*

---

> ### Author Response · Authors · 2025-11-21
>
> *(continued from the previous comment)*
>
> ---
>
> **W4-1, Q3: KLENT with MCTS**
>
> Thank you for asking a question on the performance of the KLENT-trained network with test-time MCTS. First, let us summarize our experimental results in Figure 11 in the following table.
>
> | Algorithm used for network training | Win rate without test-time search | Win rate with test-time search of 800 simulations | Difference   |
> | ----------------------------------- | --------------------------------- | ------------------------------------------------- | ------------ |
> | KLENT                               | **29%**                           | **89%**                                           | **+60 %pt.** |
> | Gumbel AlphaZero                    | 2%                                | 37%                                               | +35 %pt.     |
>
> We consider the results show that the KLENT-trained agent can significantly improve its performance with test-time search. In addition, we would like to clarify that equation $V^\pi(s) = \sum_a \pi(a|s)Q^\pi(s, a)$ is not a heuristic, because it is based on the policy $\pi$ we actually use for rollout. Based on these reasons, we also consider that the use of this equation does not limit the performance of KLENT with search.
>
> To clarify this point, we have revised the second paragraph of Appendix H. Thank you again for your thorough and detailed review.
>
> **W4-2, Q4: Framing of Contributions**
>
> Thank you for raising a question on the framing of our contributions. As the reviewer has kindly pointed out, our goal is to improve the sample efficiency of training. This sense of purpose is based on the fact that while search-based methods such as AlphaZero achieve strong performance, their significant computational demand has been pointed out as their limitation (Zhao et al., 2022). As we have discussed in our conclusion section, while our results do not preclude the effectiveness of search-based approaches, including AlphaZero when massive computational resources are available, we consider our simple and efficient approach to be valuable for most practitioners and researchers in the community.
>
> To further clarify this point, we have revised second paragraph of Section 6 in our paper. Thank you for your kind comments improving the paper.
>
> **References**
>
> * Dengwei Zhao et al. Efficient learning for AlphaZero via path consistency. ICML 2022.
>
> ---
>
> We thank the reviewer again for the helpful feedback and please let us know if there are any additional concerns or questions.

---

### Official Review · Reviewer_bWVB · 2025-10-26

**Soundness:** 3
**Presentation:** 3
**Contribution:** 3
**Rating:** 6
**Confidence:** 4

**Summary:**

The paper explores policy regularization problem for deep reinforcement learning algorithms applied to board games. Authors propose a KLENT algorithm with an actor-critic architecture that uses KLD-regularization and entropy regularization simultaniously. Proposed algorithm demonstrates state-of-the-art performance on board games including Go, Chess, Shogi and other games, outperforming AlphaZero and classic DRL algorithms.

**Strengths:**

- KLENT shows state-of-the-art performance on board games benchmark, outperforming AlphaZero and classic DRL algorithms.
- The proposed KLENT algorithm is relatively simple which allows for further improvements and application of key ideas in the future research
- A solid research on hyperparameter sensitivity for proposed algorithm.

**Weaknesses:**

- Ablation study is not conducted well enough. While it is present in the paper, there are no direct comparision of KLENT with ablated features and KLENT without ablation.

**Questions:**

**Questions:**
- Why did you choose board games as only benchmark and limit your study to it? It seems that your proposed algorithm can be applied to any environment that is suitable for actor-critic algorithms, as it do not rely on anything specific to board games.
- The algorithm seems to be similar to Soft Actor-Critic algorithm with additional KL-divergence regularization. Is there anything else aside from KLD regularization thet differs KLENT from Soft Actor-Critic? If not, it would be better to describe KLENT as SAC modification rather then an entirely novel algorithm.

**Suggestions:**
- It would be good to briefly describe in introduction or background section how algorithms designed for board games can generalize to some real-world applications. Such description will pinpoint an importance of this class of algorithms.
- In section 3.1 you describe various regularizations that were used in previous works, in which you claim that PPO uses only KL regularization, which is not correct. KL-regularized PPO is only one of two variants (another one is [PPO with clipping](https://arxiv.org/pdf/1707.06347) surrogate). Also, PPO uses entropy regularization for both its variants, which is also described in original paper. Please, reffer to this, to carefully check algorithm description provided in section 3.1.
- In figure 2 captions "state value" and "action value" is a bit confusing. It's better to change them to "value function" and "Q-function" respectively as it is aligns more with commonly used terminology.

---

> ### Author Response · Authors · 2025-11-21
>
> We thank the reviewer for the thorough review and constructive feedback about this work. We consider that the reviewer has kindly raised a weakness, questions, and suggestions on the following topics.
>
> * Weakness
>     * W1: **Ablation Study**
> * Questions
>     * Q1: **Benchmark Choice**
>     * Q2: **Differences from SAC**
> * Suggestions
>     * S1: **Connection to Real-World Applications**
>     * S2: **Explanation on PPO**
>     * S3: **Caption of Figure 2**
>
>
> We would like to answer the above as follows.
>
> **W1: Ablation Study**
>
> Thank you for raising a discussion on our ablation study. Let us briefly summarize the experimental settings of our ablation study in Section 5.2. We have compared the following five algorithms.
>
> 1. Full KLENT (= KLENT without ablation)
> 2. KLENT without entropy regularization (α=0)
> 3. KLENT without KL regularization (β=0)
> 4. KLENT without λ-returns (λ=0, λ-returns are replaced by 1-step return.)
> 5. KLENT without λ-returns (λ=1, λ-returns are replaced by Monte Carlo return.)
>
> Comparing these five algorithms, we have investigated the effect of each core technique used in KLENT. The experimental results have shown the importance of each core technique.
>
> We hope the above clarification resolves the reviewer's concern about our ablation study. If you still have any concerns or questions, please let us know, and we are willing to answer them.
>
> **Q1: Benchmark Choice**
>
> Thank you for asking a question about our benchmark choice. Our central research question is whether we can develop an efficient model-free RL algorithm compared to search-based methods such as AlphaZero. To answer this question, we consider the board game domain to be one of the best domains because of the following reasons.
>
> * Rigor rule: Board games have rigor rules and a clear objective, namely, winning the game.
> * Enough complexity: Board games are highly complex decision-making tasks, as even human experts spend minutes to hours deliberating on a single action.
> * Best testbed to compare with search-based methods: This domain is where search-based methods such as AlphaZero originated and is demonstrating state-of-the-art performance.
>
> Therefore, we considered that conducting extensive experiments for performance comparisons and ablation studies in this domain is primarily important to answer our research question.
>
> **Q2: Differences from SAC**
>
> Thank you for asking about the difference between KLENT and Soft Actor-Critic (SAC), aside from KL divergence regularization. We consider that the main difference lies in the following.
>
> 1. Difference on action spaces: SAC, which was originally developed for robotic control, mainly focuses on realizing robust learning on MDPs with continuous action spaces. On the other hand, KLENT mainly focuses on finite action spaces, which is one of the characteristics of board games. This finite action space assumption also enables obtaining an analytically optimal solution for the policy optimization problem, as in Equation 3, resulting in a different algorithm from SAC.
> 2. Difference in value learning: In addition, there is a difference in the Q-function learned. In SAC, the agent learn a Soft Q-function, which is defined as the expected cumulative sum of environmental reward and policy entropy. This makes agents prefer a more stochastic policy even if the environmental rewards are the same. On the other hand, KLENT learn an ordinary Q-function, which is defined as the expected cumulative sum of just environmental rewards, enabling direct maximization of the winning probability.
> 3. Difference between off-policy and on-policy: Furthermore, there is a difference between off-policy and on-policy. SAC is an off-policy algorithm. It sweeps the replay buffer several times, reducing the number of samples collected from the real world. On the other hand, KLENT is designed as an on-policy algorithm, aiming to achieve stable learning and reduce computational cost for training.
>
> Considering the differences above, we have described KLENT as an independent algorithm from SAC, rather than a modification of it.
>
> In response to your kind review, we consider that this point should be clearly explained in our paper. Therefore, we have added the last paragraph in Section 4.1 to explain the differences above. The paragraph also includes the comparison between KLENT and other prior studies, based on the comments by Reviewer 9LfE and Reviewer iDUE. Thank you for your helpful question.
>
> ---
>
> *(to be continued in the next comment)*

---

> > ### Author Response · Authors · 2025-11-21
> >
> > *(continued from the previous comment)*
> >
> > ---
> >
> > **S1: Connection to Real-World Applications**
> >
> > Thank you for your kind suggestion. In this study, we have focused on "board games", but as defined in Section 2.2, we have only assumed that the environment is an MDP with a finite action space. This class of problems covers several real-world applications such as discrete optimization, algorithmic discovery, and mathematical proving (Fawzi et al., 2022; Mankowitz et al., 2023; Hubert et al., 2025), and search-based methods such as AlphaZero are widely used in this domain. Our efficient algorithm may also achieve efficient learning in these practical domains, accelerating research on such real-world problems.
> >
> > As the reviewer has kindly mentioned, it is important to explain these explanations in the paper. Therefore, we have revised our introduction to include the discussion above. Thank you for your helpful suggestion for improving the paper.
> >
> > **S2: Explanation on PPO**
> >
> > Thank you so much for pointing out the explanation of PPO in detail. In response to the reviewer's suggestion, we have referred to the original paper of PPO and confirmed that we have to modify our explanation. As the reviewer has kindly pointed out, PPO has two variants of the KL-regularized one and clipped one, and also utilizes an entropy bonus to enhance exploration. Following these facts, we have revised our explanation on PPO in Section 3.1. Thank you again for pointing it out.
> >
> > **S3: Caption of Figure 2**
> >
> > Thank you for your kind suggestion. In response, we have revised the caption of Figure 2 to use the suggested terminologies of "value function" and "Q-function". Thank you for your suggestion to improve the paper.
> >
> > **References**
> >
> > * Fawzi et al., Discovering faster matrix multiplication algorithms with reinforcement learning. Nature 2022.
> > * Mankowitz et al., Faster sorting algorithms discovered using deep reinforcement learning. Nature 2023.
> > * Hubert et al., Olympiad-level formal mathematical reasoning with reinforcement learning. Nature 2025.
> >
> > ---
> >
> > We thank the reviewer again for the helpful feedback and please let us know if there are any additional concerns or questions.

---

### Official Review · Reviewer_9LfE · 2025-10-30

**Soundness:** 3
**Presentation:** 4
**Contribution:** 1
**Rating:** 4
**Confidence:** 4

**Summary:**

This paper proposes a model-free reinforcement learning algorithm KLENT, which uses actor-critic archiecture. For training of the actor it uses both KL divergence and entropy regularization and for the training of the action-value critic it uses the $\lambda$-return. The authors admit that very similar algorithms have been proposed before and aim to empirically show that they work well in the domain of board games. The authors then compare their algorithm in 5 different games with several strong baselines including AlphaZeros trained actor. Moreover, authors provide ablation study to show the effect of each KLENT component.

**Strengths:**

* The policy update rule introduced in 4.1 is novel to the best of my knowledge and could be applied in more algorithms for decision-making that use KL divergence regularization (possibly outside perfect information games).
* The performance of KLENT is improvement over strong baselines like PPO and Gumbel AlphaZero (without rollouts in test-time).
* The detailed experimental section including ablation study and test on a large game (19x19 Go).
* When combined with MCTS test-time search, KLENT could be used to replace AlphaZeros training part.

**Weaknesses:**

The main algorithm is almost identical to [1] and [2], which are applicable to broader class of games. The main difference to those algorithms seem to be the new policy-update rule and that KLENT does not use the regularization policy update from [2]. However, the paper does not explain the relationship in detail nor does it provide direct empirical comparison.

Compared to [2], which is off-policy, KLENT works only on-policy. However, the extension to off-policy KLENT seems plausible.

The strongest part of AlphaZeros performance is the additional test-time search (as evidenced by [3]). Even though KLENT could use MCTS in the test-time like the AlphaZero, which authors do in appendix H, it does break one of the strong points raised by the authors, namely "using simpler model-free algorithm to achieve similar performance as AlphaZero". I believe that KLENT would not be able to outpeform AlphaZero if AlphaZero used test-time search (and KLENT did not), given sufficient number of rollouts. Adding MCTS to KLENT could yield better performance than AlphaZero, but at the cost of being as complex as AlphaZero (possibly more complex, since the training and testing would be vastly different). I believe the paper would be stronger if it was formulated as an alternative to full AlphaZero, including search, with much better performance (due to cheaper training).

The KL regularization changes the equilibrium in the new regularized game. So it is not clear how far from the optimal strategy KLENT converges. I suppose KLENT would converge to some Quantal Response Equilibrium as in [1].


[1] Samuel Sokota, et. al. A unified approach to reinforcement learning, quantal response equilibria, and two-player zero-sum games. In Deep Reinforcement Learning Workshop NeurIPS 2022, 2022.

[2] Julien Perolat, et. al. From Poincaré Recurrence to Convergence in Imperfect Information Games: Finding Equilibrium via Regularization. Proceedings of the 38th International Conference on Machine Learning, in Proceedings of Machine Learning Research 2021, 2021.

[3] Ti-Rong Wu, Ting-Han Wei, and I-Chen Wu. Accelerating and improving alphazero using population based training. In Proceedings of the AAAI Conference on Artificial Intelligence, volume 34,pp. 1046–1053, 2020.

**Questions:**

How difficult is it to make KLENT off-policy? Is it as simple as replacing n-step $\lambda$-return with Retrace?

Based on the similarities with [1] and [2], which are designed for imperfect information games, how difficult would it be to adapt the KLENT to imperfect information games?

Have you run KLENT with any smaller examples to verify KLENT converges to the optimal strategy?

In all of the games the training was ran for more than 350M simulator evaluations. How many networks were trained with each algorithm and what was the time spent on the training?

---

> ### Author Response · Authors · 2025-11-21
>
> We thank the reviewer for the thorough review and constructive feedback about this work. We consider that the reviewer has kindly raised weaknesses and questions on the following topics.
>
> * Weaknesses
>     * W1: **Relation to Sokota+ and Perolat+**
>     * W2: **Extension to Off-policy Settings**
>     * W3: **Comparison of KLENT and AlphaZero**
>     * W4: **Convergence Limit of KLENT**
> * Questions
>     * Q1: **Extension to Off-policy Settings**
>     * Q2: **Extension to Imperfect Information Games**
>     * Q3: **Convergence Limit of KLENT**
>     * Q4: **Trained Networks and Required Time**
>
> We would like to answer the above as follows.
>
> **W1: Relation to Sokota+ and Perolat+**
>
> Thank you for mentioning important prior studies of Sokota et al. (2022) and Perolat et al. (2021). The algorithmic differences to them are the following.
>
> * Differences from Sokota et al. (2022): Our study and Sokota et al. (2022) share the idea of KL and entropy regularized policy optimization. However, the policy update rules in their practical algorithms are different. Sokota et al. (2022) modify the policy update of PPO by replacing forward KL regularization with reverse KL regularization. On the other hand, KLENT proposes a policy update rule that analytically solves the policy optimization problem as in Equation 3 of our paper.
> * Differences from Perolat et al. (2021): We consider that there are mainly two differences in the maximization objective and policy update rule. For the maximization objective, Perolat et al. (2021) propose a novel reward transformation to find the Nash equilibrium, while KLENT simply maximizes the terminal reward $R_T$, which indicates the game outcome. For the policy update rule, Perolat et al. (2021) utilize replicator dynamics, while KLENT proposes an analytical policy update rule.
>
> As the reviewer has kindly pointed out, these prior studies are closely related, and the differences from them are important to be explained. Therefore, we have revised the last paragraph of Section 4.1 to clarify these differences. The paragraph also includes the comparison between KLENT and other prior studies, based on the comments by Reviewer bWVB and Reviewer iDUE. Thank you for your helpful comments.
>
> **W2, Q1: Extension to Off-policy Settings**
>
> Thank you for raising a question of the extension of KLENT to off-policy settings. In favor of the reviewer's suggestion, we consider that KLENT can be extended to off-policy settings by replacing λ-returns with off-policy counterparts, such as Retrace(λ).
>
> To clarify this point, we have revised the last paragraph of Section 4.2 in the paper. Thank you for your insightful question.
>
> **W3: Comparison of KLENT and AlphaZero**
>
> Thank you for comprehensively reviewing our experimental results. While we acknowledge that KLENT without test-time search underperforms AlphaZero with unlimited test-time search, we consider that most of the community is interested in the comparison with fixed test-time computational resources. Therefore, we have reported both performances of reactive policies and policies with test-time search in Figures 5 and 11, respectively. In both settings, we have confirmed that KLENT achieves stronger performance compared to other approaches.
>
> For simplicity, we consider that the elimination of MCTS during training significantly reduces the implementation and development costs. For MCTS-based methods such as AlphaZero, we have to debug both the MCTS part and the RL part to achieve meaningful training. This inherent complexity of the MCTS+RL algorithm is pointed out as a barrier, especially for those new to decision intelligence (Niu et al., 2023). On the other hand, KLENT does not require MCTS during training, significantly reducing the debugging cost for training. Even if we implement test-time MCTS, we can debug RL for training and MCTS for test in separate, it is easier than debugging them at the same time. To sum up, we consider that KLENT's elimination of MCTS during training significantly reduces the overall development cost. In response to your kind review, we have added the explanation on this point in the second-to-last paragraph of Section 4.3. Thank you for your insightful review.
>
> ---
>
> *(to be continued in the next comment)*

---

> > ### Author Response · Authors · 2025-11-21
> >
> > *(continued from the previous comment)*
> >
> > ---
> >
> > **Q2: Extension to Imperfect Information Games**
> >
> > Thank you for asking a question on the extension of KLENT to imperfect information games. To adapt KLENT to imperfect information board games, we can formulate the problem as a Partially observable MDP (POMDP) instead of an ordinary MDP, and use corresponding policy and value functions in the algorithm. Specifically, we can extend KLENT to imperfect information games by modifying the state term $S_t$ in Algorithm 1 to a history term $\{(O_{t'}, A_{t'}, R_{t'})\}_{t'=0}^t$, which is a sequence of observation, action and rewards, and change the architecture of policy and value networks to take this sequence as their inputs.
> >
> > To clarify how to extend KLENT to imperfect information games, we have revised the last paragraph of Section 4.3 in the paper. Thank you for raising your insightful question.
> >
> > **W4, Q3: Convergence Limit of KLENT**
> >
> > Thank you for asking a question on the asymptotic behavior of KLENT. As the reviewer has kindly pointed out, we consider that KLENT converges to Quantal Response Equilibria (QRE). To verify this hypothesis, we have run KLENT on a small game where we can analytically compute the QRE. We have provided the results of our additional experiments in Appendix L of the revised version. The results suggest that KLENT converges to QRE, as the reviewer has correctly anticipated. We consider these results to be insightful in understanding the behavior of KLENT and mentioned it also in the first paragraph of Section 5. Thank you so much for your kind suggestion.
> >
> > **Q4: Trained Networks and Required Time**
> >
> > We trained one neural network with a ResNet-style feature extractor for each method. Policy, state-value, and action-value heads were added as required by each method. The detail of the network architecture is provided in the third paragraph of Appendix C. For GPU hours, we have spent approximately 2,000 GPU hours on NVIDIA A100 GPU in total to run the main experiments in Figures 5 to 8. We have revised the first paragraph of Appendix K to explain the computational requirements in detail. Thank you for your helpful question.
> >
> > **References**
> >
> > * Yazhe Niu, et al. LightZero: A Unified Benchmark for Monte Carlo Tree Search in General Sequential Decision Scenarios, NeurIPS 2023.
> >
> > ---
> >
> > We thank the reviewer again for the helpful feedback and please let us know if there are any additional concerns or questions.

---

> > > ### Comment · Reviewer_9LfE · 2025-11-26
> > >
> > > I would like to thank the reviewers for their comprehensive answers and changes to the paper. Some of the changes were beneficial, but I am not sure whether that is the case for all of them.
> > >
> > > I would like to start with the Weakness 3. I think authors misunderstood my point. I agree that KLENT without search outperforms AlphaZero without test-time search. I also agree that KLENT with test-time search outperforms AlphaZero with test-time search. If the paper clearly states this contribution, I would be happy to increase my score. However, that is not what the paper claims. The authors focus on KLENT being a "simple model-free alternative to AlphaZero". KLENT without search is arguably simpler than AlphaZero, but it does not have competitive performance. On the other hand, KLENT with search performs better than AplhaZero, but it is not simpler, since it requires implementing MCTS, training of the neural networks, and n-step return, which is not used in AlphaZero.
> > >
> > > I do not think this is undermines the significance of KLENT, but the current focus on the method being simpler is misleading and counterproductive. In particular, I disagree with the addition at the end of section 4.3, which claims that adding search to KLENT is easier than implementing AlphaZero.
> > >
> > >
> > > Regarding answer to Question 2:
> > > It is definitely possible to do this naive extension to imperfect information games. However, I am not sure whether this naive extension does not break some convergence guarantees. In imperfect information games, single obervation corresponds to the same state. Converging in those environments to some reasonable solution depends on the sampling probability of those different states. The authors do not reflect this in their newly added paragraph in Section 4.3. It is possible that their approach will still converge to QPE in imperfect information games with this rule, but as of right now, this is unclear and the revision seems to suggest that KLENT will work well even in imperfect information setting, which is not guaranteed. I think this change should either be better supported or taken out of the paper.
> > >
> > > As a result I will keep my score for now.

---

> ### Author Response · Authors · 2025-11-27
>
> We are deeply grateful for your extensive review and the kind additional comments for improving the paper.
>
> **W3: Comparison of KLENT and AlphaZero**
>
> Thank you for providing kind responses. Initially, we were considering that KLENT could be a simpler alternative to AlphaZero, but we have changed our minds after reading your kind and detailed response. As the reviewer has kindly pointed out, we should focus on our empirical contributions rather than the implementation cost. Our experimental results show that KLENT achieves higher win rates than search-based approaches under the same test-time computational budget, for both policies with and without test-time search. Therefore, based on this discussion, we have decided to revise the paper as follows.
>
> - Revise the last paragraph of Section 5.1 to clearly explain our empirical contributions based on the experimental results.
> - Remove the second-to-last paragraph in Section 4.3, which claims that adding search to KLENT is easier than implementing AlphaZero.
> - Revise other sentences that claim the simplicity of KLENT, such as those in the abstract and the introduction.
>
> Aside from the above points, we would like to note that we will release our code for the experiments in public if the paper is accepted. We believe sharing our code would enhance the reproducibility of our results and contribute to the research community. Thank you again for your kind comments on improving the paper.
>
> **Q2: Extension to Imperfect Information Games**
>
> We are also grateful for your detailed response. As the reviewer has kindly pointed out, the naive extension we have explained is possible, but it is unclear whether this extension maintains the convergence of KLENT to an equilibrium. In order to make the paper more evidence-based and solid, we have decided to take out this paragraph from the paper.
>
> ---
>
> Following your kind comments, we have updated our paper. If you have any remaining concerns, we would appreciate it if you could let us know. We would be happy to address them. Thank you again for your kind and detailed review.

---

### Official Review · Reviewer_r3T6 · 2025-11-05

**Soundness:** 4
**Presentation:** 4
**Contribution:** 2
**Rating:** 4
**Confidence:** 3

**Summary:**

This paper gives a simple model-free RL method for board games that eliminates look-ahead tree search during training. The motivation is to reduce the heavy computational requirements of search-based approaches like AlphaZero. The method uses a policy optimization approach with lambda returns.

**Strengths:**

The paper is clearly written, and reports convincing performance gains in learning efficiency. KLENT achieves higher win rates or faster training progress than heavy search-based algorithms (AlphaZero, Gumbel-AlphaZero) under the same computational budget. Environment and baseline is comprehensive, and hyper-parameter and implementation details are given. A key strength is the algorithm’s simplicity relative to AlphaZero-style methods. By avoiding MCTS, the method is much easier to implement and requires less specialized data structure.

**Weaknesses:**

It is important to note that the paper’s value may lie in the empirical finding that this straightforward combination works remarkably well on complex board games. Demonstrating that “model-free RL (with proper regularization) can rival search-based methods in these games” is a useful result for the community, especially for those who cannot afford massive search-based training. However, the lack of algorithmic novelty means the paper’s contributions are primarily empirical and engineering-oriented. To strengthen the paper as a research contribution, the authors would need to either provide new insights arising from this combination or propose an innovative modification. Currently, the success of the method, while notable, can be attributed to known techniques applied diligently rather than to a creative new idea introduced by the authors. That is, beyond the combination of existing very common (and nowadays standard) tricks, the approach does not introduce a fundamentally new mechanism or theory.

The related work discussion could be stronger in distinguishing this work from prior efforts. There have been other attempts at applying deep RL to board games (the authors cite AlphaZero alternatives and some resource-reduced versions). However, the paper does not cite any older approach that tried model-free learning in these games – possibly because prior to AlphaZero, non-search RL (like TD-Gammon for backgammon, or Atari methods applied to simpler board games) existed.

**Questions:**

A little speculation: the authors might consider whether a small amount of search at test-time could further boost the performance?

---

> ### Author Response · Authors · 2025-11-21
>
> We thank the reviewer for the thorough review and constructive feedback about this work. We consider that the reviewer has kindly raised weaknesses and a question on the following topics.
>
> - Weaknesses
>     - W1: **Novelty**
>     - W2: **Related Work**
> - Question
>     - Q1: **Performance Boost with Test-Time Search**
>
> We would like to answer the above as follows.
>
> **W1: Novelty**
>
> Thank you for raising a discussion on the novelty of this work. Our main contribution is showing that a simple model-free algorithm can achieve efficient learning than search-based methods such as AlphaZero through extensive experiments on board games. Therefore, we acknowledge that our contributions are empirical rather than algorithmic.
>
> However, we consider our empirical results to be indeed novel to the community, as search-based methods are considered to be state-of-the-art in this domain. Furthermore, as the reviewer has kindly mentioned, we consider that our simple and efficient approach is valuable for most of the community, especially for those who cannot afford massive computational resources. In response to your review, we have revised the second-to-last paragraph of the introduction section to properly explain our contributions.
>
> For theoretical properties, we expect KLENT to converge to a quantal response equilibrium (QRE; McKelvey & Palfrey, 1995), a softened version of an optimal strategy. To examine this, we conducted additional experiments on an analytically solvable small-scale game and report the results in Appendix L. The results suggest that KLENT indeed converges to the QRE, providing empirical support for this expectation and helping to clarify the theoretical properties of KLENT.
>
> We hope that these revisions help alleviate the concerns the reviewer has kindly raised regarding the novelty of this study. If you have any remaining concerns, we would appreciate it if you could let us know. We would be happy to address them. Thank you for your comprehensive review.
>
> **W2: Related Work**
>
> Thank you for raising a discussion on prior work. In the initial version of our paper, we cited TD-Gammon only in the introduction section, therefore we have revised the paper to properly cite it also in the related work section. In addition, in response to your kind review, we have conducted an additional survey on prior efforts before AlphaZero and found the following papers.
>
> * Schraudolph et al. (NIPS 1993): Tried TD learning to learn the position evaluation function in 9x9 Go.
> * Thrun (NIPS 1994): Used TD learning and neural networks to learn to play Chess.
> * Baxter et al. (ACNN 1998): Proposed TDLeaf(λ), which combines TD learning with minimax tree search.
> * Van der Ree and Wiering (ADPRL 2013): Compared the performances of TD-learning, Q-learning, and SARSA in Othello.
>
> We have revised Section 3 of our paper to cite these prior efforts in order to strengthen our related work discussion. Thank you for your helpful comments.
>
> **Q1: Performance Boost with Test-Time Search**
>
> Thank you for raising a question about the performance boost of the proposed method KLENT with test-time search. We have conducted experiments to investigate it and included results in Appendix H. Our results in Figure 11 show that the KLENT can improve its performance with test-time search. Therefore, we consider the reviewer's speculation to be correct.
>
> To further clarify these points, we have revised the last paragraph of Section 5.1. Thank you for your helpful question.
>
> **References**
>
> - McKelvey & Palfrey, Quantal Response Equilibria for Normal Form Games. Games and Economic Behavior 1995.
> - Nicol N. Schraudolph et al. Temporal Difference Learning of Position Evaluation in the Game of Go. NIPS 1993.
> - Sebastian Thrun. Learning to Play the Game of Chess. NIPS 1994.
> - Jonathan Baxter et al. TDLeaf(λ): Combining Temporal Difference Learning with Game-Tree Search. ACNN 1998.
> - Michiel van der Ree and Marco Wiering. Reinforcement Learning in the Game of Othello: Learning Against a Fixed Opponent and Learning from Self-Play. ADPRL 2013.
>
> ---
>
> We thank the reviewer again for the helpful feedback and please let us know if there are any additional concerns or questions.

---

### Author Response · Authors · 2025-11-25
**Remarks by Authors**

We would like to thank all the reviewer for their thoughtful and comprehensive reviews. We have uploaded the latest version of our paper. In this official comment, let us summarize the discussion so far.

For the strengths of this study, all reviewers have kindly highlighted the good learning efficiency of the proposed method KLENT, and have evaluated our experiments as extensive. In addition, the reviewer r3T6, vsPq, and iDUE kindly mentioned that the paper is clearly written and easy to follow. The reviewer r3T6 and bWVB have also mentioned the algorithmic simplicity of KLENT as its strength. Thank you for your kind and thoughtful reviews.

The reviewers 9LfE, vsPq, and bWVB have raised questions on the algorithmic difference of the proposed method KLENT from existing approaches (Haarnoja et al., 2018; Grill et al., 2020; Perolat et al., 2021; Sokota et al., 2023), and we have explained that the use of an analytically optimal policy in KLENT is the common algorithmic differences from them.

The reviewers r3T6, vsPq, and iDUE have kindly raised discussions on the contributions, and we have explained that the main contributions of this study lie in the following points.

* The experimental results demonstrate that properly revisiting existing techniques can achieve efficient learning in the board game domain with a model-free RL.
* The ablation studies show that the proper combination is important to achieve high performance.
* In response to Q3 by Reviewer 9LfE, we have conducted additional experiments on a small-scale game and discussed the convergence behavior of KLENT.

We consider these empirical results to be original and valuable to the ICLR community due to the following reasons.

* Demonstrating the strength of model-free RL in the board game domain is novel, as search-based methods such as AlphaZero have long been believed to be state-of-the-art in this domain.
* Our efficient approach can be a useful alternative to search-based methods, especially for those new to decision intelligence and who cannot afford massive computational resources.

We have revised our introduction section to clarify the discussion above.

**Revisions:** Reviewers also kindly provided comments, questions, and suggestions on other topics.  We have responded to each comment and revised the paper as shown in the following table. Revisions are written in red in the paper.

| Topic | Corresponding Review  | Our Response |
| --- | --- | --- |
| Contributions and novelty of the study | W1 by Reviewer r3T6, W1 by Reviewer vsPq, W2 by Reviewer iDUE | Revised the second-to-last paragraph of Section 1.|
| Potential real-world applications | S1 by Reviewer bWVB  | Added the last paragraph of Section 1.|
| Explanation of PPO  | S2 by Reviewer bWVB | Revised Section 3.1.|
| Classical approaches for board games| W2 by Reviewer r3T6 | Revised Section 3.1 and Section 3.2.  |
| Differences between KLENT and existing algorithms | Q2 by Reviewer bWVB, W1 by Reviewer 9LfE, Q1 by Reviewer iDUE | Added a third paragraph to Section 4.1.|
| Terminologies in caption| S3 by Reviewer bWVB | Revised the caption of Figure 2. |
| Extention of KLENT to off-policy settings | W2 by Reviewer 9LfE | Revised the last paragraph of Section 4.2.|
| Performane boost with test-time search | Q1 by Reviewer r3T6| Revised and the last paragraph of Section 5.1. |
| Comparison of KLENT and AlphaZero  | W3 by Reviewer 9LfE| Revised and the last paragraph of Section 5.1 and other sentences. |
| Convergence limit of KLENT | Q3 by Reviewer 9LfE, W1 by Reviewer r3T6 | Conducted additional experiments and revised first paragraph of Section 5 and added Appendix L. |
| Performance comparison to Grill+  | Q1 by Reviewer iDUE | Conducted additional experiments and updated Figures 1 and 5. |
| The number of collected data  | Q3 by Reviewer iDUE | Revised second paragraph of Section 5.1.  |
| Evaluation protocol | W3 by Reviewer vsPq, Q2 by Reviewer iDUE | Added a fourth paragraph to Section 5.1 and Table 7 in Appendix H.  |
| Framing of Contributions| Q4 by Reviewer vsPq | Revised second paragraph of Section 6.|
| KLENT with test-time MCTS| Q3 by Reviewer vsPq | Revised second paragraph of Appendix H.|
| Total GPU hours| Q3 by Reviewer iDUE | Revised first paragraph of Appendix K.|

**References**

* Haarnoja, et. al. Soft Actor-Critic: Off-Policy Maximum Entropy Deep Reinforcement Learning with a Stochastic Actor. ICML 2018.
* Grill, et al. Monte-Carlo tree search as regularized policy optimization. ICML 2020.
* Perolat, et. al. From Poincaré Recurrence to Convergence in Imperfect Information Games: Finding Equilibrium via Regularization. ICML 2021.
* Sokota, et. al. A unified approach to reinforcement learning, quantal response equilibria, and two-player zero-sum games. ICLR 2023.

---

If you have any further concerns or questions, we would appreciate it if you could let us know. Once again, we are so grateful to the kind and detailed comments provided by the reviewers.

---

### Author Response · Authors · 2025-12-02
**Official Comment by Authors to Area Chair**

Dear Area Chair,

Thank you very much for reviewing our paper. During the rebuttal period, the five reviewers kindly raised a number of comments and questions, and we have responded to all of them in our rebuttal. We believe that our responses have addressed the reviewers’ concerns and that the resulting revisions have significantly improved the paper. A summary of these discussions and revisions is provided in the comment below titled “Remarks by Authors.”

We appreciate your time and effort in evaluating our submission.

Sincerely,
The authors of paper #12682

---

### Public Comment · ~Kazuki_Ota1 · 2026-07-03
**Update: Revised Version Accepted to ICML 2026**

We are happy to share that a substantially revised version of this work has been accepted to ICML 2026 as “Revisiting Regularized Policy Optimization for Stable and Efficient Reinforcement Learning in Two-Player Games.”

Compared with the ICLR submission, the revised version reframes the method as regularized policy optimization for two-player games, expands the experiments with test-time search, and adds theoretical analysis.

For details, please see:
* Project page: https://kazukiohta.github.io/klent/
* Paper: https://arxiv.org/abs/2602.10894
* Code: https://github.com/KazukiOhta/klent

---

### Meta-Review · Area_Chair_6uMY · 2025-12-12

**Summary:**

This paper proposes a simple model-free RL algorithm that demonstrates remarkably superior performance in board games. The policy learned from this model-free RL algorithm, when paired with MCTS, outperforms AlphaZero with MCTS on 9 x 9 Go games, suppose the same computation budget is given. All reviewers appreciate the remarkable performance improvement.

My evaluation is below. First, I ignored all the concerns about lacking of algorithmic innovation or theoretical analysis. Given the very remarkable performance improvement in a wide range of board games, the simplicity is actually an advantage. However, I do completely agree with 9LfE that the current story is completely wrong and misleading. From a scientific perspective, using AlphaZero without MCTS as a baseline makes no sense to me because everything of AlphaZero is designed with MCTS in mind. Although without MCTS, AlphaZero still performs reasonably well, it deviates a lot from the original design of AlphaZero. This is also a point made by vsPq but unfortunately I do not find a direct response to this point.

I quote from 9LfE
> The authors focus on KLENT being a "simple model-free alternative to AlphaZero". KLENT without search is arguably simpler than AlphaZero, but it does not have competitive performance. On the other hand, KLENT with search performs better than AplhaZero, but it is not simpler, since it requires implementing MCTS, training of the neural networks, and n-step return, which is not used in AlphaZero.

I completely agree with this evaluation. The most important message in this submission is "KLENT with search performs better than AplhaZero". I suggest the authors to take the comment from 9LfE very seriously and significantly revise the paper, particularly, all statements of "model-free" should be removed or toned down.

The authors did make a good start for such revision and promised changes in their response to 9LfE. But I envision the necessary revision would be really large and to support the new narrative, they need to replicate Figure 11 on more games.

In the authors final remark, it is stated that
>Demonstrating the strength of model-free RL in the board game domain is novel, as search-based methods such as AlphaZero have long been believed to be state-of-the-art in this domain.

I believe this is wrong for the reasons above. Particularly, search-based methods ARE still SOTA even given KLENT. It is KLENT+search that outperforms AlphaZero (by default, when we say AlphaZero, MCTS is included).

>Our efficient approach can be a useful alternative to search-based methods

It is not really clear to what extent KLENT is a useful alternative.

I acknowledge the empirical significance but the wrong conclusion in this final author remark makes me doubt whether the camera-ready version will have a correct narrative if accepted. So I have to recommend reject.

**Reviewer Concerns:**

See the summary above.

**Reviewer Scores:**

r3T6 may not change. 9LfE may increase to 6. bWVB may not change. vsPq may not change. iDUE may not change.

---

### Decision · Program_Chairs · 2026-01-26

Reject